

# Bipolar carbon and hydrogen isotope constraints of the Holocene methane budget

Jonas Beck[1,2], Michael Bock[1,2], Jochen Schmitt[1,2], Barbara Seth[1,2], Thomas Blunier[3], Hubertus Fischer[1,2]

[1]Climate and Environmental Physics, Physics Institute, University of Bern, Sidlerstrasse 5, 3012 Bern, Switzerland
[2]Oeschger Centre for Climate Change Research, University of Bern, 3012 Bern, Switzerland
[3]Centre for Ice and Climate, Niels Bohr Institute, University of Copenhagen, Juliane Maries Vej 30, 2100 Copenhagen, Denmark

*Correspondence to*: Jonas Beck (beck@climate.unibe.ch)

**Abstract.** Atmospheric methane concentration shows a well-known decrease over the first half of the Holocene following the northern hemisphere summer insolation before it started to increase again to preindustrial values. There is a debate about what caused this change in the methane concentration trend, in particular, whether an early anthropogenic influence or natural emissions led to the reversal of the atmospheric $CH_4$ concentration. Here, we present new methane concentration and stable hydrogen and carbon isotope data measured on ice core samples from both Greenland and Antarctica over the Holocene. With the help of a two-box model and the full suite of $CH_4$ parameters, the new data allow us to quantify the total methane emissions in the northern and southern hemispheres separately as well as their isotopic signatures, while interpretation of isotopic records of only one hemisphere may lead to erroneous conclusions. For the first half of the Holocene our results indicate a decrease in northern and southern hemisphere $CH_4$ emissions by more than 30 Tg $CH_4$/yr in total accompanied by a drop in the northern carbon isotopic source signature of about −3‰. This cannot be explained by a change in the source mix alone, but requires shifts in the isotopic signature of the sources themselves caused by changes in the precursor material for the methane production. In the second half of the Holocene global $CH_4$ emissions increased by about 30 Tg $CH_4$/yr, while preindustrial isotopic emission signatures remained more a less constant. However, our results show that the increase of methane emissions starting in the mid- Holocene took place in the southern hemisphere, while northern hemisphere emissions started to increase only about 2000 years ago. Accordingly, natural emissions in the southern tropics appear to be the main cause of the $CH_4$ increase starting 5000 years ago in contradiction to an early anthropogenic influence on the global methane budget by East Asian land use changes.

## 1 Introduction

Atmospheric methane ($CH_4$) is a potent greenhouse gas and its concentrations are strongly coupled to the Earth's climate system. Due to the human influence on the Earth system, the $CH_4$ concentration ($[CH_4]$) in the atmosphere has increased by a factor of 2.5 (relative to the preindustrial level) over the last centuries and contributes significantly to the human-induced radiative forcing (Dlugokencky et al., 2005; Etheridge et al., 1998). Today both anthropogenic $CH_4$ sources (rice agriculture, livestock, fossil fuel production, anthropogenic biomass burning and landfills) and natural $CH_4$ sources (natural wetlands, wildfires, geologic emissions, wild animals (including termites) and marine $CH_4$ hydrates) contribute to the global $CH_4$ emissions (Kirschke et al., 2013). The main mechanism removing $CH_4$ from the atmosphere is the chemical reaction of $CH_4$ with OH radicals in the troposphere. Together with the other sink processes (such as stratospheric loss, reaction with Cl radicals in the marine boundary layer and soil uptake) the OH sink determines the recent atmospheric life time of $CH_4$, which is between 8.4 and 10.6 years (Naik et al., 2013).

Despite the fact that the total $CH_4$ emissions today are well known, there is still considerable debate about the individual contributions of (in particular small) $CH_4$ source and their variability. Moreover, a long-standing debate exists whether long-





term Holocene $CH_4$ trends are naturally caused by wetland emissions (Singarayer et al., 2011) or whether an anthropogenic influence started already 4-5000 years ago by early human land-use changes (Ruddiman, 2003) (see below). Accordingly, to better assess the human influence on the global methane cycle, robust estimates of the natural (pre-anthropogenic) $CH_4$ emissions and their climate-coupled variations are required. Only polar ice cores can provide this information as they represent

a direct, albeit low-pass filtered archive of trapped air and, therefore, offer the possibility to investigate the atmospheric composition of the past. With such ice core measurements the millennial to centennial evolution of the atmospheric $[CH_4]$ has been determined back to 800 thousand years before present (ka BP), where present refers to the year 1950, from the Antarctic EPICA Dome C (EDC) ice core showing glacial/interglacial variations in atmospheric $[CH_4]$ by a factor of about 2 (Loulergue et al., 2008). More important for the assessment of the human influence on $CH_4$ levels are their variations during the Holocene,

i.e. our current interglacial that started about 10,000 years ago. In this time period $[CH_4]$ decreased by about 100 parts per billion (ppb) until about 5000 yr ago (Figure 1) after an early maximum during the preboreal period (about 11.5-10 ka BP) (Blunier et al., 1995; Brook et al., 1996; Flückiger et al., 2002; Schilt et al., 2010). $CH_4$ values started to slowly increase again in the mid-Holocene until the fast anthropogenic increase resulting from the industrialisation starting around 200 years ago (MacFarling Meure et al., 2006). High resolution $CH_4$ measurements on the West Antarctic Ice Sheet (WAIS) and Greenland

GISP2 (Greenland Ice Sheet Project 2) ice cores (Mitchell et al., 2013) show also centennial $CH_4$ variations of 20-30 ppb over the last 2800 years superimposed on the long-term increase.

With the trend reversal in the mid-Holocene leading to a "bowl shape" (Sowers, 2010) of $[CH_4]$, the Holocene differs from many previous interglacials, where the $[CH_4]$ steadily dropped to glacial levels concurrently with the northern hemisphere

summer insolation. A controversial hypothesis postulates that $CH_4$ emissions caused by early human land use were responsible for the $CH_4$ turnaround about 5 ka BP (Ruddiman, 2003; Ruddiman and Thomson, 2001). According to these authors, early farming activities, mainly rice agriculture in the eastern part of China, led to significant releases of $CH_4$ long before the industrial era (Ruddiman et al., 2008) and, thus, the authors propose that the "bowl" was shaped by human activities. Another explanation supported by a model-based study explains the late Holocene $CH_4$ rise by calling upon increased emissions from

southern tropical wetlands due to an insolation-driven strengthening of the monsoon in the western Amazon (Singarayer et al., 2011). Accordingly, this model is able to produce a global $[CH_4]$ rise in the Holocene driven by changes in the orbital forcing only. Note that due to the different orbital configuration during the last interglacial (MIS5.5), this model is also able to reproduce the steadily declining $[CH_4]$ during MIS5.5, when strongly declining boreal $CH_4$ emissions overcompensate increasing southern tropical wetland emissions.

An important fact is that the two scenarios explain the Holocene $CH_4$ rise with additional emissions taking place in different geographic regions. By measuring the $[CH_4]$ on ice cores from both polar regions (Greenland and Antarctica), the inter polar difference (IPD) can be calculated. This quantity can be used to draw conclusions about the hemispheric distribution of the $CH_4$ emissions (Brook et al., 2000; Chappellaz et al., 1997; Mitchell et al., 2013; Yang et al., 2017). Another tool to investigate

the processes involved in the $CH_4$ cycle are measurements of the stable isotope signature in ice core $CH_4$. The $CH_4$ released by the various sources are associated with typical hydrogen and carbon isotopic signatures $\delta D$-$CH_4$ (D representing Deuterium or $^2H$) (Quay et al., 1999; Walter et al., 2008; Whiticar, 1993; Whiticar and Schaefer, 2007) and $\delta^{13}C$-$CH_4$ (Etiope et al., 2008; Quay et al., 1999; Walter et al., 2008; Whiticar, 1993; Whiticar and Schaefer, 2007). In addition, the sink processes lead to a strong fractionation (Feilberg et al., 2005; Levine et al., 2011b; Snover and Quay, 2000; Whiticar and Schaefer, 2007), which

leave atmospheric $CH_4$ being strongly enriched in heavy isotopes relative to the isotopic emission signatures. Changes of the relative contributions in sources lead to an alteration of the atmospheric $CH_4$ isotopic composition. $CH_4$ isotope measurements on ice cores, therefore, allow us to deduce the role the different processes related to the $CH_4$ cycle played in the past. By measuring both $\delta D$-$CH_4$ and $\delta^{13}C$-$CH_4$ from the Greenland GISP2 ice core over the Holocene, Sowers (2010) was able to



provide first evidence of possible changes in the $CH_4$ precursor material due to a shift in the C3/C4 plant ratio, but could not unambiguously answer the question about the Holocene $CH_4$ anomaly.

In this study we combined for the first time comprehensive information on the $CH_4$ IPD and measurements of the carbon and hydrogen isotopic signature of $CH_4$ by measuring all three parameters – the $[CH_4]$, $\delta D\text{-}CH_4$ and $\delta^{13}C\text{-}CH_4$ – on ice samples from both polar regions with a high analytical precision and improved resolution. Furthermore, we use a two-box model approach, which allows us to deconvolve the atmospheric signal, to calculate the total emissions and their isotopic signatures in each hemisphere.

As mentioned above, changes in the sinks leave an imprint on the isotopic composition of $CH_4$ in the atmosphere as well. However, the $CH_4$ sinks are thought to scale proportionally with the source (not changing the $CH_4$ life time) and only play a minor role for the observed atmospheric $[CH_4]$ variation since the last glacial maximum (Levine et al., 2011a; Martinerie et al., 1995). Therefore, the discussion of this study centres around $CH_4$ source processes, hence emissions of different $CH_4$ source categories.

## 2  Method

### 2.1  Measurements

The stable isotope data ($\delta D\text{-}CH_4$ and $\delta^{13}C\text{-}CH_4$) presented in this study were measured at the University of Bern on samples from the NGRIP (North GReenland Ice core Project) ice core from Greenland. Antarctic samples from the EDML (EPICA Dronning Maud Land) and the TALDICE (TALos Dome Ice CorE) ice cores were analysed for $\delta D\text{-}CH_4$ and $\delta^{13}C\text{-}CH_4$,
respectively (Bock et al., 2017). Note that for both parameters the samples from Greenland and Antarctica have been measured during the same time interval with the same measurement system by the same operators and in randomised order. This procedure is crucial to avoid any systematic error (e.g. due to a long-term drift of the system) and ensures unbiased results in our source deconvolution.

The two isotopic parameters were measured at the University of Bern on two independent measurement systems designed for this specific purpose as described in detail in previous publications (Bock et al., 2010; Schmitt et al., 2014). They follow the same principal procedure: the air is (i) extracted from the ice sample by purging the melt water, (ii) water vapour, $CO_2$ and the major air compounds ($N_2$, $O_2$ and Ar) are removed from the gas sample, (iii) $CH_4$ is separated from other trace gases using gas chromatography (GC), (iv) $CH_4$ is pyrolysed to $H_2$ or combusted to $CO_2$ in the $\delta D$ and $\delta^{13}C$ systems, respectively, and (v) the
isotopic composition of $H_2$ and $CO_2$ is measured using an isotope-ratio mass spectrometer (IRMS). For both systems effects of interfering masses, specifically Krypton, in the mass spectrometer are avoided by technical adaptations (Schmitt et al., 2013). In the case of $\delta D\text{-}CH_4$, the focussed $H_2$ peak is separated from other components using a second GC separation (post-pyrolysis trapping) (Bock et al., 2014). In the $\delta^{13}C\text{-}CH_4$ system, the $CH_4$-derived $CO_2$ is captured and delayed using a fused silica trap (at $-196°C$) after the combustion oven and thereby separated from Krypton (Schmitt et al., 2014). Note that many
$\delta D\text{-}CH_4$ and $\delta^{13}C\text{-}CH_4$ values based on such a gas chromatography - mass spectrometry method reported in the literature are not corrected for this effect, leading to significant offsets especially for low $CH_4$ concentrations.

The $\delta D\text{-}CH_4$ values are determined relative to our primary air standard 'Air Controlé', a recent clean air ($[CH_4]$ = 1971 ± 7 ppb), which has been cross-referenced to $-93.6 ± 2.2 ‰$ with respect to (wrt) Vienna Standard Mean Ocean Water (VSMOW)
using 'Alert 2002/11', an air collected at Alert Station, Canada (Bock et al., 2014). The latter was referenced by Poss (2003) to be $-82.15 ± 0.3 ‰$ wrt VSMOW, which anchors our $\delta D$ values to the isotope scale of the University of Heidelberg (Bock





et al., 2010, 2014). Note that so far no internationally uniform δD-CH$_4$ standardisation scale has been established leading to systematic interlaboratory offsets as described in Umezawa et al. (2017). The measurement precision (1σ) of 2.3 ‰ was derived by calculating the pooled standard deviation of multiple NGRIP replicate (vertically neighbouring samples) measurements. The reference gas for the δ$^{13}$C-CH$_4$ measurements is 'Boulder', an ambient air diluted with CH$_4$-free air ([CH$_4$]

= 1508.18 ± 0.17 ppb) calibrated by the National Oceanic and Atmospheric Administration (NOAA) to a δ$^{13}$C-CH$_4$ value of −47.34 ± 0.02 ‰ with respect to Vienna Pee Dee Belemnite (VPDB) (Schmitt et al., 2014; Umezawa et al., 2017). The measurement precision (1σ) derived from replicate measurements is better than 0.15 ‰ for ice core samples.

### 2.2 Monte Carlo spline approximation of millennial CH$_4$ changes

To compare values from different ice core datasets (with different and variable temporal resolutions and measurement precisions) interpolation of the data to a common age scale is needed. Additionally, the records have to be low-pass filtered to the same (millennial) time scales that can be resolved in the isotope records. Therefore, in this study smoothing splines following Enting (1987) are used to calculate a continuous low-pass filtered record from the discretely measured values. The stiffness of the spline curve is characterised by its cutoff period. A Python (http://www.python.org/) routine was developed to

calculate splines in a Monte Carlo manner, where the data points are varied within a normal distribution within their standard deviation defined by the individual uncertainty of the measured values. This procedure allows for the calculation of the average of the splines representing the best guess mean evolution of the data on the time scale resolved by the spline. Furthermore, using the standard deviation of all splines, the spread of the individual splines can be translated to an uncertainty of the mean spline. This spline calculation is used for various purposes in this study. The choice of the cutoff period has a large influence

on the outcome of the spline and is, therefore, always documented wherever it is selected manually. For some applications (e.g. in the routines described in the sections 2.3.2 and 2.3.3), however, we apply an objective rule to define an appropriate cutoff period for a data set: wherever the cutoff period is not specified explicitly, it is set to eight times the median of the data resolution.

If the sampling rate in some parts of a dataset is much higher than in others, this part has a larger impact on the spline than other coarser resolved sections. For short periods of large signal change (e.g. the 8.2 ka event for CH$_4$) such an oversampling relative to other parts of the dataset may lead to significant artefacts in the calculated spline. Wherever this might be an issue, an additional step was introduced into the spline calculation routine, where the data were down-sampled in a randomised way within the Monte Carlo loop (i.e. for each individual spline) to end up with a sub set of a prescribed number of data points,

where the chance of data points to get picked are inversely proportional to the local data density. The down-sampled data are still unequally distributed, but the local average data density is similar over the whole time period studied.

### 2.3 Data correction and compilation

For further analysis of our new data, we complemented our [CH$_4$] records with data from other studies from the GRIP (GReenland Ice core Project), GISP2, TALDICE, EDML, WAIS (West Antarctic Ice Sheet) and EDC ice cores shown in

Table 1. However, combining different datasets introduces various challenges. In the following paragraphs we describe how these difficulties have been addressed.

### 2.3.1 Time synchronisation

All ice cores have individually determined gas age scales, which allow us to assign an age to each sample. Since we compare values from different ice cores (and from both hemispheres) small deviations in the age calculations have significant

consequences for the interpretation (e.g. artefacts in the deconvolution (see section 2.4)). In an attempt to synchronise the





different datasets, we linearly stretched and compressed the individual age scales between manually-chosen tie points in three steps:

1. Wherever possible, we tied the data sets of each of the cores to the same reference age scale. For CH$_4$ datasets that reach far
enough to the present, the fast CH$_4$ increase of the last centuries was used to synchronise the time scales. As a reference we used [CH$_4$] data from Law Dome firn air measurements (MacFarling Meure et al., 2006) and an atmospheric [CH$_4$] history derived from NEEM firn air (Buizert et al., 2012) representing the CH$_4$ evolution in Antarctica and in Greenland, respectively. Both records cover the entire anthropogenic CH$_4$ increase and overlap with recent atmospheric measurements (CSIRO, 2017; Dlugokencky et al., 2017). For the early Holocene the layer-counted NGRIP δ$^{18}$O-H$_2$O record was used as an absolute time
marker. Where the CH$_4$ data show strong signals such as at the 8.2 ka event and the Younger Dryas-Holocene transition these were synchronised to the corresponding δ$^{18}$O signal, assuming that there is no significant lag between the temperature changes recorded in δ$^{18}$O and the response of CH$_4$ source to this change (Baumgartner et al., 2014).

2. With the use of splines of different cutoff periods and their time derivatives we were looking for typical features and signals
in the [CH$_4$] evolution of the different data sets within the same hemisphere to define additional tie points and synchronise the data sets for each hemisphere. However, within both hemispheres the temporal alignments of all the distinguishable CH$_4$ features were already within the limitation of the data resolution. Therefore, no additional tie points could be defined at this stage.

3. In the third synchronisation step we aligned the data of both hemispheres after applying the offset corrections (see section 2.3.2). Additional time corrections (for the southern hemisphere data) could be defined by looking at the fine structure of the [CH$_4$] evolution (e.g. the mid-Holocene [CH$_4$] minimum (around 5400 a BP) or the local [CH$_4$] maximum before the early Holocene decrease (around 9850 a BP)). Over the late Holocene period Mitchell et al. (2013) provided already synchronised data for the GISP2 and the WAIS ice core based on decadal to centennial variability; we retained their time scale over this
period.

Table 2 lists the original age scales of the different ice cores. The tie points (original age and age correction) used to achieve an optimal temporal synchronisation between the CH$_4$ datasets are shown in Table 3. Temporal shifts to data with ages younger than the first tie point and older than the last tie point were chosen to be identical to those of the first and last tie points, respectively. Please note that for the ice cores, where the primary age scales provide an age uncertainty, all but three tie point
corrections (GRIP: 273 a; TALDICE: 127 a and 6042 a) lie within the 1$\sigma$-range of the uncertainty given for the individual age scales.

### 2.3.2 Offset correction

Another issue that can complicates compiling multiple datasets are concentration offsets. The CH$_4$ datasets we use to
complement our new data were measured with different methods, in different labs, and at different times over the last decades. If we use the information about the relative evolution of the methane concentration documented by other measurement series, one has to be aware that the absolute level might show an offset to our data due to calibration and extraction issues. Therefore, we correct all datasets for such potential offsets. To do so, we calculated residuals defined as the difference between the spline values of the CH$_4$ data from other labs evaluated at the age of our data and the CH$_4$ mixing ratios of our data. A least-squares
optimisation routine was used to find the offsets for the spline values of the different data sets. The offset corrections applied to the data are shown in Table 1. The GISP2 and WAIS datasets by Mitchell et al. (2013) are treated as a single dataset, as



they were measured in the same lab and during the same measurement campaign, and their difference (Greenland-Antarctica) represents the true IPD. In Figure 1 the synchronised and offset-corrected $CH_4$ time series as used in this study are shown.

### 2.3.3 Outlier detection

The GISP2 datasets from Brook et al. (1996) and Brook (2009) show an increased incidence of anomalously high $[CH_4]$ values
(relative to other data sets) in the mid-Holocene. As this ice is from the brittle ice zone (the zone where air bubbles and clathrates coexist in situ, and which is prone to bad core quality due to damage during the pressure and temperature relaxation process after the core was brought to the surface) this is probably due to infiltration of modern air into the ice samples in (micro) cracks opened in the ice after core retrieval (Gow et al., 1997; Neff, 2014). To avoid an artificial increase of the splined signal, which would result in a significant blow-up of the IPD, we identify and remove such outliers within the depth interval
650-1400 m of the GISP2 ice core. To do so we used an iterative elimination approach. In each step, first the two GISP2 datasets were corrected for offsets relative to each other (Brook et al. (1996) was shifted to fit with Brook (2009) by 36 ppb) and combined to one data set. Thereafter, for each GISP2 data point the difference between its value and the spline of all other GISP2 data in the brittle zone was calculated and divided by the $1\sigma$ uncertainty of the Monte Carlo spline at its age. If the largest (positive) relative residuum exceeded the threshold value of 4, the corresponding data point was removed. This
procedure was repeated until no outliers could be found any more. Table 4 shows the data points that were identified and eliminated through this method (also shown in Figure 1). In total 15 out of 95 data points (within this depth range) were removed, reflecting the bad core quality in this depth interval. Note that we make the assumption here that the $[CH_4]$ evolution in the atmosphere is relatively smooth. As we focus only on millennial changes in the data here, this assumption does not affect our conclusions.

### 2.3.4 Isotope data corrections

All stable isotope data have been corrected for gravitational settling. This process takes place in the firn column and leads to an enrichment of the heavier isotopes at the depth where the air bubbles are finally closed off and the gas is trapped in the ice (Schwander, 1996). Using $\delta^{136}Xe$ data, which come as a side product during the $\delta^{13}C$-$CH_4$ measurement of the Bern $\delta^{13}C$ system, and additional published $\delta^{15}N$-$N_2$ data (Capron et al., 2013; Eggleston et al., 2016; Landais et al., 2006), we
experimentally derived this gravitational enrichment and directly correct the $CH_4$ stable isotope data, as this gravitational enrichment is only dependent on the mass difference between the two isotopes considered. The correction for the TALDICE $\delta^{13}C$ values increases from the present to the early Holocene from 0.28 ‰ to 0.39 ‰. The NGRIP $\delta^{136}Xe$ data do not show a systematic signal. Thus, a constant value of 0.32 ‰ is subtracted from the NGRIP isotope values. Only very few $\delta^{15}N$-$N_2$ data points in the early Holocene were available to correct the EDML $\delta D$-$CH_4$ record. Since in the case of $\delta D$-$CH_4$ the measurement
error and the observed signal are much larger than the gravitational enrichment, it is adequate to assume that this correction is also constant over time without any consequences for the interpretation of the data. For the EDML $\delta D$-$CH_4$ record the correction of 0.46 ‰ was applied.

Due to the relatively constant climatic conditions and no rapid $[CH_4]$ changes occurring over the Holocene, thermal diffusion
(Severinghaus et al., 1998) and diffusive fractionation (Buizert et al., 2013) – other processes potentially leading to an artefact in the stable isotope signature – are not considered in this study.

### 2.4 Deconvolution model

Our deconvolution model is a tool to disentangle the different processes involved in the atmospheric methane cycle. More
precisely, it allows us to obtain quantitative information about the emission processes that led to the local tropospheric



composition at the ice core sites in the past. In other similar studies different approaches of varying complexity and with different boundary conditions have been presented (Baumgartner et al., 2012; Fischer et al., 2008; Mitchell et al., 2013). For this study we use a two-box model where the boxes represent the northern and southern tropospheric hemispheres.

The following six deconvolution equations allow us to use the six measured parameters (the tropospheric [CH4] ($c_x$) and the two stable isotope ratios ( $^iR_x$)) for each hemisphere to calculate the emissions ($E_x$) and their isotope ratios ( $^iR_{E_x}$) for both boxes analytically (with $x$ and $y$ representing north and south and $i$ the mass numbers 2 or 13 for deuterium and $^{13}$C):

$$E_x = m^* \cdot \left( \frac{r_x \cdot c_x}{\tau_x} + \frac{c_x}{2 \cdot \theta} - \frac{c_y}{2 \cdot \theta} \right) \quad (1)$$

$$^iR_{E_x} = \frac{m^*}{E_x} \cdot \left( \frac{r_x \cdot c_x}{\tau_x} \cdot {}^iR_x \cdot {}^i\alpha_x + \frac{c_x}{2 \cdot \theta} \cdot {}^iR_x - \frac{c_y}{2 \cdot \theta} \cdot {}^iR_y \right) \quad (2)$$

The first term in equations Eq. (1) and Eq. (2) represents the loss due to the CH4 sinks, and the other two terms represent the hemispheric mixing (out of and into the box, with $\theta$ being the mixing time between the hemispheres as explained in more

detail below). The parameter $m^* = \frac{m_0}{c_0}$ translates atmospheric concentrations (ppb) to total atmospheric inventories (in Tg) using the mean atmospheric concentration value $c_0 = 1650\ ppb$ of the year 1987 and the corresponding global CH4 burden $m_0 = 4800\ Tg$ (Steele et al., 1992). The transport between the boxes is quantified by the exchange time $\theta = 1.56\ a$, which is determined through a calibration using recent sulphur hexafluoride (SF6) data (described in section 2.4.1).

Note that the sizes of the two well-mixed boxes are not equal; the sizes are defined by the global mean annual latitude of the inter-tropical convergence zone (ITCZ) $\varphi_{ITCZ} = 5°N$ (Marshall et al., 2014), which acts as a barrier to the hemispheric exchange of air masses. The inequality of the box sizes is reflected in Eq. (1) and Eq. (2) by

$$r_x = \frac{V_x}{V_{atm}} = \frac{1}{2} \left( 1 \mp \sin\left( \varphi_{ITCZ} \right) \right), \quad (3)$$

the ratio of the box volume relative to the total atmospheric volume. The global mean life time of CH4 quantifies the total methane sink and is set to $\tau = 8.4\ a$ (as in the 4-box model used in Fischer et al. (2008) and in Bock et al. (2017)). According

to the relative importance given in Kirschke et al. (2013) (average of bottom-up estimates of all three time intervals) the sink is proportionally divided into the four different sink processes ($s_{OH}, s_{strat}, s_{soil}, s_{Cl}$) shown in Table 5. While $s_{OH}$ and $s_{strat}$ are thought to be equally distributed north and south of the ITCZ and, therefore, are set equal in the two boxes, $s_{soil}$ and $s_{Cl}$ are partitioned according to the area ratios of land and open water, respectively, derived from an 0.5°-by-0.5° resolution land-cover database (Channan et al., 2014). The life time of CH4 in each box

$$\tau_x = \tau \cdot \frac{\sum s_{tot}}{\sum s_x} \cdot r_x, \quad (4)$$

is determined by the north-south distribution of the sinks and the relative box sizes.

The different sink processes lead to strong fractionations in both isotopes (Feilberg et al., 2005; Houweling et al., 2000; Quay et al., 1999; Snover and Quay, 2000 and references therein) and yield the hemispheric isotope fractionation factors $^i\alpha_x = 1 + {}^i\varepsilon_x$, with the fractionations $^i\varepsilon_x$ shown in Table 5. As in our model with only two tropospheric boxes $s_{strat}$ is

applied to the total tropospheric CH4 inventory instead of the isotopically more depleted stratospheric CH4, the literature values for the stratospheric sink fractionation have to be adjusted to $^2\varepsilon_{strat} = -170\ ‰$ and $^{13}\varepsilon_{strat} = -13.1\ ‰$. These values have been empirically derived to match the tropospheric isotopic signature of the forward 4-box model described by Fischer et al. (2008) which contains two stratospheric boxes. First, using the identical CH4 emission setup in both models, the north-south air mass exchange of the 2-box model has been increased to equal the tropospheric [CH4] values with those of the 4-box



model (IPD). In a second step, the fractionations of the stratospheric sink $^i\varepsilon_{strat}$ in the 2-box model have been adjusted to match the tropospheric δD-CH₄ and δ¹³C-CH₄ values of the two models.

As the individual data points from ice core [CH₄] and isotope measurements are subject to uncertainty in both time and values even after above homogenisation steps, we only use the millennial variations in CH₄, δD-CH₄ and δ¹³C-CH₄ in our deconvolution model. To this end we calculated low-pass filtered versions of the measured data using the Monte Carlo spline approximations described in section 2.2, which allow us to quantify the mean and the uncertainty of the millennial variations. The uncertainty of the splines of the tropospheric values are transmitted to the emission values using Gaussian error propagation. Due to the mentioned limitations we refrain from analysing shorter-scale variability.

### 2.4.1 SF₆ calibration of the two box model

The same approach used to set up the CH₄ deconvolution described in the previous section (Eq. (1) and (2)) allows us to formulate the general equations for the change of the concentration of any trace gas species in the two hemisphere boxes:

$$\frac{dc_x}{dt} \cdot r_x \cdot m^* = E_x - S_x - \frac{m^*}{2 \cdot \theta}\left(c_x - c_y\right) \tag{1}$$

with $\frac{dc_x}{dt}$ representing the change rate of the hemispheric concentration, $m^*$ the translation factor linking a concentration to an atmospheric inventory, $E_x$ and $S_x$ the emissions and the sinks in Box $x$, and $r_x$ the ratio of the box volumes depending on the mean annual latitude of the ITCZ $\varphi_{ITCZ}$ as described above, and $x$ and $y$ standing for north and south (or vice versa).

SF₆ is a strong greenhouse gas with a life time of more than 1000 years and is solely of anthropogenic origin mainly emitted in the northern hemisphere ($r_{E_n} = 95\,\%$, $r_{E_s} = 5\,\%$) (Geller et al., 1997, p.6; Kovács et al., 2017; Levin et al., 2010). NOAA/ESRL (2017) measurements taken at Alert Station (Canada), Summit (Greenland), Palmer Station (Antarctica) and South Pole (Antarctica) record the SF₆ evolution in both polar regions over the last decades. Since the target in this study is the long-time trend and not seasonal variations, the NOAA/ESRL data are smoothed using a smoothing spline with a cutoff period of 2 years. For every year in the recorded period (1996-2008 CE) the annual SF₆ increase ($\frac{dc_x}{dt}$) and the mean concentration ($c_x$ and $c_y$) are calculated. The records of the global emissions ($E_{tot}$), the global atmospheric mean SF₆ concentration ($c_{tot}$) and the total atmospheric inventory ($m_{tot}$) published by Levin et al. (2010) are used to calculate the hemispheric emissions $E_x = r_{E_x} \cdot E_{tot}$ and the translation factor $m^* = mean\left(\frac{m_{tot}}{c_{tot}}\right) = 25.7\,\frac{Gg\,SF_6}{ppt}$. With the approximation of a negligible sink term (since life time exceeds the time scale of interest by 2-3 orders of magnitude) Eq. ((1) can be solved for the hemispheric exchange time:

$$\theta = \frac{m^*\left(c_x - c_y\right)}{2\left(\frac{dc_x}{dt} \cdot r_x \cdot m^* - r_{E_x} \cdot E_{tot}\right)} \tag{2}$$

Using Eq. (2) the exchange time can be calculated for each year and hemisphere. With $\varphi_{ITCZ} = 5.0°N$ we get the average value of $\theta = 1.56 \pm 0.17\,a$.

In reality the air is not perfectly mixed within each hemisphere, which means that our ice core measurements do not perfectly represent the hemispheric average. However, by using the high latitude SF₆ data for the model calibration, we inherently correct for the fact that a signal subject to the inter-hemispheric exchange has to be transported to the poles before it is captured in the ice. Accordingly, $\theta$ has to be interpreted as an interpolar rather than an interhemispheric exchange time.





### 2.4.2 Deconvolution of artificial time series

The interpretation of the output of our inversion model is challenging and not always intuitive. To get a better understanding of the model output, some simplified experiments using artificial time series have been performed and are shown in this study for didactic reasons. In a first approach all six parameters are held constant over time. In the panels on the left in Figure 2 the

tropospheric [$CH_4$] (a), and the isotopic signatures $\delta D\text{-}CH_4$ (b) and $\delta^{13}C\text{-}CH_4$ (c) are shown for the northern box in red and for the southern box in blue. The constant values (solid lines) represent the levels of the splines of our measurements at 2 ka BP and serve as input to the deconvolution model. The panels on the right show the deconvolution results, i.e., the integrated emissions (d) and their mean isotopic signatures (e, f) that are emitted into the two hemisphere boxes according to our model. As the input parameters are constant, the emission signal calculated by the deconvolution to achieve the prescribed

tropospheric values must be constant in all parameters as well. However, although the northern hemisphere concentrations are only 6 % larger than the southern ones, a large difference in the emission strengths ($E_n \approx 2.2 \cdot E_s$) is needed to maintain the IPD in the [$CH_4$] compensating the air mass exchange between the atmospheres. Note also that due to the sink fractionation the isotopic signature of atmospheric $CH_4$ is strongly enriched in the heavy isotopes compared to the emissions (see Table 5).

For the second experiment (dashed lines in Figure 2), the IPD of the tropospheric isotope signatures ($IPD_{\delta D}$ and $IPD_{\delta 13C}$) have been changed. We kept the southern values constant, while we linearly changed the northern values so that the minimal and the maximal $IPD_{\delta D}$ and $IPD_{\delta 13C}$ values match the splined isotope data, respectively (dashed lines in (b) and (c)). This leads to changes in the calculated isotope signatures of the emissions, whereas the emission strengths are unaffected. As shown (dashed lines) in panels (e) and (f) the isotopic signature for both the northern and the southern hemisphere emissions need to be altered

to yield the given atmospheric values (of constant [$CH_4$] and isotopic signatures changing only in one hemisphere). To achieve the declining isotope signal in the northern troposphere, the isotope signatures of emissions from the northern hemisphere have to become isotopically more depleted in heavy isotopes (further on conveniently referred to as isotopically "lighter", equivalent to lower values δ-values) over time. At the same time the emissions in the southern hemisphere compensate for the isotopically lighter $CH_4$ mixed down from the northern box and, therefore, become more enriched in the heavy isotope (isotopically

"heavier", equivalent to higher δ-values) over time. Since the southern source is much weaker than the northern source (more northern-sourced $CH_4$ is mixed from north to south than southern-sourced $CH_4$ is mixed from south to north), the change in the isotopic signatures of the emissions turns out to be even larger in the southern hemisphere.

The dotted line in panel (a) in Figure 2 shows a third scenario, where all tropospheric parameters are kept constant except for

the [$CH_4$] in the northern hemisphere. Similar to the former experiment, the [$CH_4$] in the northern box is varied between values representing the minimal and the maximal measured $IPD_{CH4}$. The dotted curves in the right panels (d-f) in Figure 2 indicate that this change has an influence on all six emission parameters of our model. The emission strength in the northern box increases to produce the rising [$CH_4$]. Declining emission strength in the southern hemisphere compensates for the increasing $CH_4$ north-to-south flux due to the rising $IPD_{CH4}$. The $CH_4$ that enters a box by atmospheric mixing is already fractionated by

the sinks and isotopically heavier than the newly emitted $CH_4$. If the relative amount of $CH_4$ input by mixing is decreased for the northern (increased for the southern) box, the emission signatures become less (more) depleted in heavy isotopes to maintain the tropospheric $CH_4$ isotopic signatures.

Since the variations for both experiments were done based on a reasonable range (observed changes in IPDs), the span for the

calculated values of the emission parameters also provides a range for the changes that are needed to alter the measured tropospheric IPDs in the Holocene. Note, however, that the variation scenarios (e.g. with a fully decoupling of atmospheric concentration and the isotopic composition of $CH_4$) do not represent changes we expect to observe in nature and are presented for didactic reasons only.



## 3 Results

All tropospheric data (this study complemented with published [CH₄] data from Blunier et al., (1995); Brook et al., (1996); Brook, (2009); Chappellaz et al., (1993, 1997); Flückiger et al., (2002); Mitchell et al., (2013); Schilt et al., (2010), see section 2.3, Figure 1 and Table 1) are shown in Figure 3 in the left-hand panels. The smoothing splines, which are all calculated with

the same cutoff period of 3000 years, are shown including their $1\sigma$ uncertainty bands. In panel (a) the composite of the time-synchronised and offset-corrected CH₄ data is shown over the Holocene. The diamond-shaped symbols represent the NGRIP and TALDICE data of this study measured during the $\delta^{13}$C-CH₄ measurement campaign. Overall, our data compilation after the above mentioned corrections supports previous trends: after high concentrations in the Preboreal (ca. 11.5-10 ka BP) the data show a decline of the atmospheric [CH₄] over the first half of the Holocene followed by a reversal of this trend and an

accelerating increase from the mid- to the late Holocene. The preindustrial level is comparable to the [CH₄] during the Preboreal warm period. The splines emphasise this long-term shape nicely but, deliberately, the splines do not reproduce the fine structure in the data (see for example Mitchell et al. (2013)). The difference between the [CH₄] values in the northern and southern hemispheres (the IPD$_{CH4}$) is also clearly visible in the data. The concentration measured on ice cores from Greenland (red) is on average 45 ppb higher than the values derived from Antarctic ice cores (blue), which agrees with the result of

Chappellaz et al. (1997).

In panel (b) the measured $\delta$D-CH₄ values show a pronounced difference of −16.3 ‰ between northern and southern values, which is rather stable over the whole period of investigation. Limited by the coarser resolution of the EDML $\delta$D-CH₄ data, the smoothed curves and the error bands illustrate the probable evolution of the true tropospheric signals only on millennial

timescales and longer. Compared to the large [CH₄] changes, the $\delta$D-CH₄ signal over the Holocene is relatively small and similar in the two datasets. The values become isotopically lighter in the first half of the Holocene and heavier again in the second half. Note that our NGRIP $\delta$D-CH₄ data differ strongly from the dataset from the GISP2 ice core published by Sowers (2010) (see Figure 4). First, there is a difference of the mean level between the two datasets of 10 to 15 ‰, which can be attributed to an inter-laboratory difference (Bock et al., 2014; Umezawa et al., 2017). Second, the GISP2 data suggested a shift

of about 20 ‰ towards heavier values from 5 to 1 ka BP, which is not confirmed by our data. This disagreement, however, may be largely attributed to the much larger scattering of the GISP2 dataset, which does not allow to quantify an unambiguous trend in $\delta$D-CH₄.

In panel (c) the $\delta^{13}$C-CH₄ measured on ice from the NGRIP and TALDICE ice cores are shown. Both datasets show a gradual

trend towards lighter values over the first half of the Holocene and almost constant values over the second half. The point of inflection is well-defined at about 5 ka BP. At about the same time the $\delta^{13}$C-CH₄ signals measured in the two hemispheres diverge from a difference of −0.35 ‰ to −0.63 ‰, which is rather constant before and after the point of inflection. Our NGRIP $\delta^{13}$C-CH₄ record is well in line with the GISP2 data by Sowers (2010) (see Figure 4), which have been subsequently corrected for Krypton interference during the measurement (Schmitt et al., 2013).

Our deconvolution model calculates the emissions that led to the measured tropospheric signals shown in the panels on the left. In panel (d) the strength of the integrated emissions (in Tg CH₄ per year) in each of the two troposphere boxes are shown. Additionally, the total CH₄ global emission (sum of north and south) are shown in grey. Note that the uncertainty ($1\sigma$) in emission fluxes is very small and much smaller than the long-term changes in the emission fluxes over the Holocene indicated

by the spline average. This uncertainty reflects only the error in emissions fluxes for changes on millennial time scales that are resolved in our spline. As indicated in high-resolution data (for example Mitchell et al., 2013) the centennial variability of atmospheric CH₄ is high and the uncertainties of reconstructed emission fluxes on such shorter time scales would be significantly higher as well.



The following observations can be made from the deconvolution results: The $CH_4$ emissions into the northern box are more than two times higher than the emissions into the southern box. After a decrease of roughly 30 Tg $CH_4$/a in the early Holocene the northern emissions remain relatively constant between 8 and 2 ka BP before increasing by 15 Tg $CH_4$/a. In the southern

hemisphere the emissions show a slight increase in the first 2,000 years of the record followed by a 22 Tg $CH_4$/a decrease until 4-5 ka BP after which emissions increase by 25 Tg $CH_4$/a in the second half of the Holocene.

The weighted averaged δD signature of all the integrated $CH_4$ emissions is shown in panel (e) for each hemisphere. With values of about $-295\,‰$ the northern emission δD signature is on average 30‰ lighter than the southern hemisphere emissions.

The δD-$E_x$ signal shows long-term variations around the Holocene mean that are in antiphase between the northern and southern hemisphere, however these variations in each hemisphere are not significantly different from their respective Holocene mean within their $2\sigma$ uncertainty. The global mean δD-E (grey line) remains essentially constant. Based on these two observations, we attribute the δD-$E_x$ fluctuations in each hemisphere to the measurement uncertainty of the individual data points of the low-resolution δD-$CH_4$ EDML record which affect the spline reconstruction.

Our experiments with artificial time series (see section 2.4.2) show that a change in the isotopic signature in one hemisphere has a large influence of different sign on the emission isotope signal in the other hemisphere (Figure 2). Hence, the absence of a long-term trend in δD-$E_x$ over the Holocene with the significant changes in the atmospheric [$CH_4$] is remarkable and a strong constraint on the $CH_4$ budget. However, the large uncertainty in δD-$E_x$ does not allow us to make robust conclusions about

millennial variations in the hydrogen isotopic signature of $CH_4$ emissions.

The $δ^{13}C$-$CH_4$ values of the emissions shown in panel (f) indicate a significant shift of the northern emission to heavier values over the Holocene, which is especially pronounced in the time interval before 4 ka BP. In the south $δ^{13}C$-$E_s$ shows also a slight negative trend over the first half of the Holocene, which however, is still well within the $1\sigma$ deconvolution uncertainty. A

stronger positive trend, however, occurs during the second half of the Holocene, which is still with the $2\sigma$ uncertainty of the reconstruction. The global mean of emissions reflects the coherent trends in the measured atmospheric values, with a strong negative trend of $-1.8‰$ over the first 5000 years of the Holocene.

### 3.1 Sensitivity studies

The size-ratio of the two boxes of the deconvolution model is defined by the annual mean latitude of the ITCZ $\varphi_{ITCZ}$. In fact,

there is evidence from different proxies of a southward migration of the ITCZ during the Holocene (Haug et al., 2001; McGee et al., 2014; Zhao and Harrison, 2012). Since it is not possible to deduce robust numbers for the position and the movement of the global mean ITCZ from local studies, $\varphi_{ITCZ}$ has been kept constant over the time at a reasonable Holocene value in our deconvolution (described in section 2.4). However, sensitivity runs have been performed to quantify the impact of this assumption on the conclusions drawn from our deconvolution results.

As described in section 2.4.1, $SF_6$ calibration has been used to determine the inter-hemispheric mixing time $\theta$ associated to different values of $\varphi_{ITCZ}$. For the first sensitivity study these two parameters have always been fitted in parallel to ensure that the different model configurations all fit the $SF_6$ constraint in an optimal way. In Figure 5 the splines of the Holocene data (left panels) and the corresponding emission values derived by the deconvolution calculations (right panels) are shown. In the

sensitivity runs, the deconvolution equations have been solved using other ($\varphi_{ITCZ}$, $\varepsilon$)-couples with $\varphi_{ITCZ}$ moving from 0.0°N to 10.0°N. In all six emission parameters, these variations only lead to small changes relative to the total signal variability. For the isotopic signatures of the emissions, all curves lay well within the $1\sigma$ error of the 5.0°N results.



A change of the mean annual position of the ITCZ has an influence on the box size distributions. This has a direct impact on the north-south partitioning of the individual $CH_4$ sinks and thus on all six deconvolution parameters. Therefore, another sensitivity run was carried out, where only $\varphi_{ITCZ}$ was varied and $\theta$ kept constant at the best guess value. As shown in Figure

5    6, the sensitivity of the deconvolution to a change of $\varphi_{ITCZ}$ is very low if $\theta$ is kept constant. Note that the southward migration of the ITCZ over the Holocene is thought to be smaller than 1° (McGee et al., 2014). The results of this second set of experiments where only one parameter was changed (i.e. the constraints of the $SF_6$ calibration were not fulfilled) also showed that on average variations in $\varphi_{ITCZ}$ and $\theta$ contribute by a similar amount to the deviation from the 5°N solutions (see Figure 6).

## 4   Discussion

The main goal of this study is to profit from the improved $CH_4$ multi-parameter dataset using our inversion model in order to evaluate the different scenarios and to constrain the global $CH_4$ budget over the Holocene. To this end, we consider the early, mid- and late Holocene separately.

### 4.1   Early Holocene (11-8 ka BP)

The significant drop in the northern emission strength together with the slight shift towards isotopically lighter carbon emissions in the northern hemisphere from 11 to 8 ka BP is challenging to account for. One potential solution to account for the observed concentration shift would be a general weakening of the boreal $CH_4$ sources as a result of the decreasing northern summer insolation (Berger and Loutre, 1991). However, this should lead to a trend in both isotope emission signatures towards heavier values in the northern hemisphere, as in general the high-latitude $CH_4$ emissions from wetlands are isotopically light

in both isotopes (Walter et al., 2008; Whiticar and Schaefer, 2007). Accordingly, this conflicts with the stable isotope results from our deconvolution. For $\delta D$-$E_n$ the deconvolution might be affected by one single measurement point (EDML $\delta D$-$CH_4$) at 8.19 yr BP forcing the emission signature curves of the two hemispheres apart and indicating that $\delta D$-$E_n$ becomes lighter. Nevertheless, even if this single data point is removed, the deconvolution signal would be constant over this period but would not show a trend toward heavier values as expected from a decline in boreal emissions.

Based on $^{14}C$-dating Walter et al. (2007) estimated the $CH_4$ emissions from thermokarst lakes to decrease from 26 to 5 Tg $CH_4$/a in the period from 11 to 8 ka BP in line with the decreasing trend in northern hemisphere emissions in our deconvolution. To match the result of the isotope deconvolution, another isotopically heavy northern source (e.g. biomass burning) has to be reduced simultaneously to compensate for the lack of isotopically light thermokarst $CH_4$ emissions. However, charcoal records

from sediments do not indicate a decline of wild fires in the northern hemisphere but rather the opposite during the Holocene (Daniau et al., 2012).

Part of the answer of this puzzle may lie in the combined temporal evolution of the strengths and isotopic signature of the boreal wetlands. Due to land ice and permafrost retreat in the early Holocene, minerotrophic fens, characterised by relatively

high $CH_4$ emissions which are not as strongly depleted in $^{13}C$, turn into ombrotrophic bogs over time with more $^{13}C$-depleted but also much weaker $CH_4$ emissions (Ding et al., 2005; Hornibrook, 2013; Yu et al., 2013). This reduction of $CH_4$ emissions in the course of conversion of fens to bogs requires a very depleted source signature of the bogs to quantitatively explain our deconvolution results and may therefore not suffice to explain the $\delta^{13}C$-$E_n$ shift towards lighter values in the early Holocene. Furthermore, an enrichment trend in $\delta D$-$E_n$ (of about 5‰) would be expected if the $\delta D$-depleted high-latitude emissions

became relatively less important, while – if at all - we see a 20‰ depletion in $\delta D$-$E_n$ from 10 to 8 ka BP. Also the isotopic



signature of the water used in methanogenesis has a direct imprint on the hydrogen isotopic signature of the emitted $CH_4$ (Whiticar and Schaefer, 2007). Speleothem $\delta^{18}O$ records confirm an isotopic shift towards lighter values in the meteoric water in the summer monsoon regions of the northern hemisphere from 11 to 8 ka BP (Wang et al., 2014) and an inverse relationship in the southern hemisphere, in line with our deconvolution. However, the speleothem records also suggest opposite trends over

the rest of the Holocene, which are not seen in our deconvolution results. In addition, isotopically light melt water from the retreating northern hemisphere ice sheets (for example in proglacial lakes) may contribute to a depletion in the hydrogen isotopic signature of boreal $CH_4$ source in the early Holocene.

### 4.2  Mid-Holocene (8-5.5 ka BP)

Our deconvolution results show that the decrease in atmospheric $CH_4$ concentrations over this interval are mainly caused by
declining southern hemisphere emissions. This decrease in the southern hemisphere $CH_4$ emission from 8 to 5.5 ka BP - during the time of the strongest increase of austral summer insolation (Berger and Loutre, 1991) - appears counterintuitive but is in line with the model results by Singarayer et al. (2011) explaining it with changes in tropical rainfall location. This is also the period when the ITCZ is thought to have started moving southward as a response to the change in the orbital forcing (Haug et al., 2001; McGee et al., 2014; Zhao and Harrison, 2012), which has an influence on the hemispheric $CH_4$ source distributions.
$CH_4$ emissions from regions that were attributed to the southern hemisphere box in the early Holocene may now contribute $CH_4$ to the northern hemisphere due to the shift of the ITCZ. Note that such a change not only has an influence on the spatial emission distribution, but also on other parameters, which are relevant for the deconvolution model, where the box sizes are kept constant over time. Our sensitivity studies (see section 3.1) show, however, that a movement of the ITCZ in the model has only a small effect on the deconvolution results.

As discussed above the $\delta D$ signal of the $CH_4$ emissions is mostly constant in both hemispheres. This is expected if one hemisphere box gains emissions from tropical wetlands, characterised by stable isotope signatures close to the integrated source mix, at the cost of the other box. The $\delta^{13}C$ shift of the emissions could be related to a shift in the ratio of C3 to C4 plants as suggested by Sowers (2010), while a further evolvement of northern fens to bogs is incompatible with the relatively constant
emissions in the northern hemisphere. Under Holocene climate conditions, with high availability of humidity and $CO_2$, C3 plants successfully compete against the relatively $^{13}C$-depleted C4 plants and thereby alter the carbon isotopic signature of the precursor material of methanogenesis (Farquhar et al., 1989). This process was also suggested by Möller et al. (2013) to explain the close $CO_2$ correlation of the $\delta^{13}C$-$CH_4$ signal and the decoupling from the $[CH_4]$ in the glacial period.

### 4.3  Late Holocene (5.5-1 ka BP)

The early anthropogenic influence hypothesis (Ruddiman and Thomson, 2001) calls upon increasing emissions from rice paddies mainly in Eastern Asia. In our box model such emissions would be located in the northern hemisphere box and, therefore, lead to an increase in northern emissions after 5 ka BP. However, the deconvolution shows no significant increase in $E_n$ before 2 ka BP. In contrast at 5.5 ka BP, at the time when all six presented tropospheric records imply a change in the global $CH_4$ cycle, the deconvolution indicates an amplification of southern hemisphere $CH_4$ emissions. This clearly supports
the idea of a natural Holocene wetland $CH_4$ emission increase as suggested by Singarayer et al. (2011). In contrast, a southern anthropogenic source can be excluded since the southern hemisphere was barely populated in the mid-Holocene (Kaplan et al., 2011). The observed shift in $\delta D$-$E_s$ and especially $\delta^{13}C$-$E_s$ towards heavier values in this interval cannot be explained simply by increased tropical wetland emissions in the Amazon but requires also an enhancement of a southern hemisphere $CH_4$ source strongly enriched in $^{13}C$. Increasing emissions from tropical and subtropical wild fires in the southern hemisphere,
which are also documented in charcoal records (Daniau et al., 2012), could have become increasingly important and would readily explain the joint information derived from deconvolution results on southern hemisphere $CH_4$ emissions and its stable



isotopic signatures. Other isotopically heavy sources such as geological emissions and marine $CH_4$ hydrates are unlikely to dramatically gain importance only in one hemisphere and can, therefore, be ruled out as important players driving the observed $CH_4$ changes.

## 5 Conclusions

The presented records of [$CH_4$], $\delta D$-$CH_4$ and $\delta^{13}C$-$CH_4$ measured on samples from polar ice cores from both hemispheres provide valuable insights into the Holocene $CH_4$ cycle. A significant IPD for $\delta^{13}C$ and $\delta D$, as already documented for [$CH_4$], exists over the entire period of investigation. While the [$CH_4$] data confirm the well-known Holocene evolution, the two stable isotope records provide additional insights. This shows the value of our multi isotope approach needed not to draw the wrong conclusions. A two-box model approach has been used to deconvolve the measured signals. The calculated hemispheric $CH_4$

emission signatures, characterised by different trends in the two hemispheres for $\delta^{13}C$ and no statistically significant variation in $\delta D$, show a decoupling of the emission signatures of the two isotopes. Therefore, we attribute the observed changes of the atmospheric $CH_4$ isotopes to shifts in the isotopic source signatures of individual $CH_4$ sources rather than to changes in the global $CH_4$ source mix.

In the early Holocene (11-8 ka BP) we associate the decline of northern hemisphere emissions and the $\delta^{13}C$ trend towards lower values with the natural evolution of high latitude wetlands from fens (high $CH_4$ emissions) to bogs (lower but strongly $^{13}C$ depleted $CH_4$ emissions). An alternative explanation calling upon a significant decrease of $CH_4$ emissions from thermokarst lakes requires compensation by other processes to fulfil the $\delta^{13}C$ constraint, e.g. a significant decline of wild fire activity in the northern hemisphere without such evidence. In the mid-Holocene (8-5.5 ka BP) the deconvolution shows a decline of the

southern hemisphere $CH_4$ emissions and an ongoing shift of the $\delta^{13}C$ emission signatures towards lighter values, which might be related to the southward migration of the ITCZ and the change of the C3-to-C4 plant ratio altering the carbon isotopic composition of the precursor material for $CH_4$ production. During the second half of the Holocene (5.5-1 ka BP), which shows a major increase in $CH_4$ emissions accompanied by a shift towards higher $\delta^{13}C$ signatures in the southern hemisphere, our results favour the 'Holocene anomaly' to be caused by natural $CH_4$ emissions (both wetlands and wildfires) in the southern

tropics rather than by early rice agriculture in east Asia.

The $CH_4$ stable isotope data provide a powerful constraint on the Holocene $CH_4$ system (e.g. to benchmark $CH_4$ emission models), however, the relatively large uncertainty ranges in the calculated emission parameters still limit our ability to draw more robust conclusions. The deconvolution would probably benefit from an increased temporal resolution in the southern

hemisphere $\delta D$-$CH_4$ record. However, since the concentration data have large influence on all six deconvolution parameters, we see the largest potential to further constrain the Holocene $CH_4$ emissions by actually improving the [$CH_4$] records. Better $IPD_{CH4}$ data measured on the same system during the same measurement campaign in high temporal resolution, which ensures an accurate synchronisation (such as the data by Mitchell et al. (2013) for the last 2,800 years), would bring us a large step forward in further constraining the Holocene $CH_4$ cycle. This is especially true for the Greenland ice cores within the brittle

ice zone (e.g. GISP2 records for depths between 650 and 1400 m).

*Data availability.* Data presented in this study are available at (URL Pangaea page)

*Acknowledgments.* The research leading to these results has received funding from the European Research Council (ERC)
under the European Union's Seventh Framework Programme FP7/2007-2013 ERC Grant 226172 [ERC Advanced Grant Modern Approaches to Temperature Reconstructions in Polar Ice Cores (MATRICs)] and the Swiss National Science Foundation. This work is a contribution to the North-GRIP ice core project, which is directed and organised by the Department of Geophysics at the Niels Bohr Institute for Astronomy, Physics and Geophysics, University of Copenhagen. It




is supported by funding agencies in Denmark (SNF), Belgium (FNRS-CFB), France (IFRTP and NSU/CNRS), Germany (AWI), Iceland (RannIs), Japan (MEXT), Sweden (SPRS), Switzerland (SNF), and the United States (NSF). Furthermore, this work is a contribution to the European Project for Ice Coring in Antarctica (EPICA), a joint European Science Foundation/European Commission scientific program funded by the European Union and national contributions from
Belgium, Denmark, France, Germany, Italy, The Netherlands, Norway, Sweden, Switzerland, and the United Kingdom. The main logistic support was provided by Institut Polaire Français Paul-Emile Victor (IPEV) and PNRA (at Dome C) and AWI (at Dronning Maud Land). The Talos Dome Ice Core Project (TALDICE), a joint European program led by Italy, is funded by national contributions from Italy, France, Germany, Switzerland, and the United Kingdom. The main logistical support was provided by Programma Nazionale di Ricerche in Antartide (PNRA) at Talos Dome.
This work is EPICA publication no. xx and TALDICE publication no. yy.

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

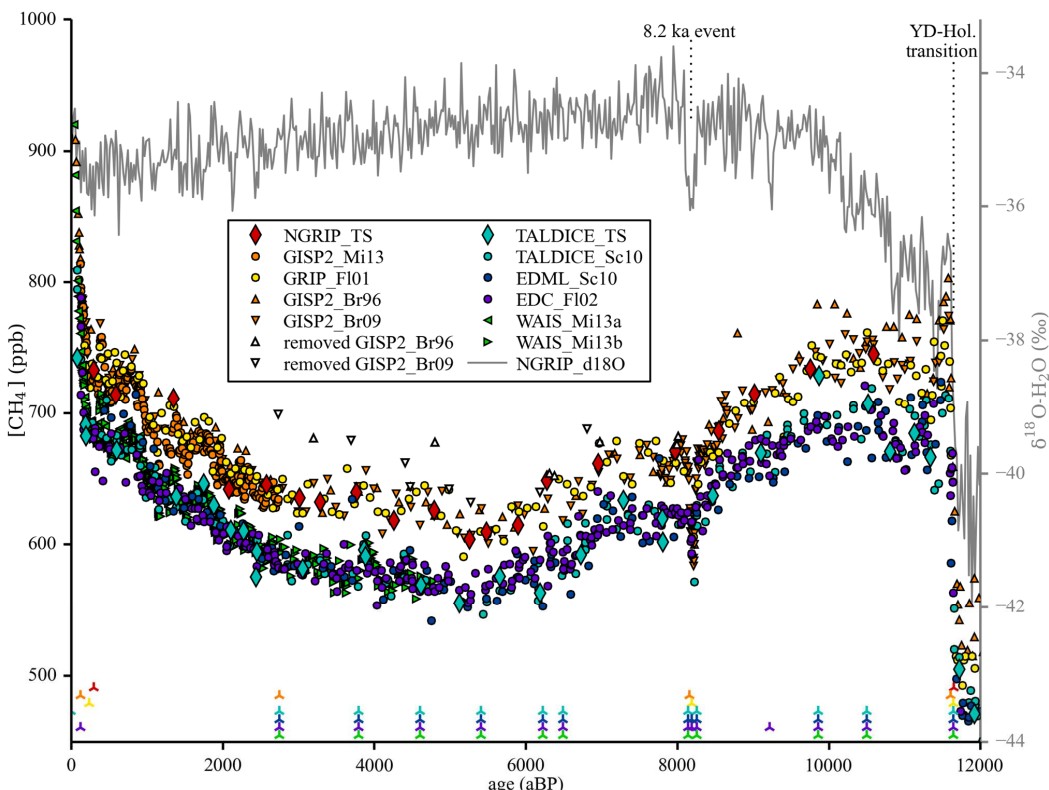

**Figure 1: CH₄ concentration data compilation.** Discrete [CH₄] data over the Holocene used for the CH₄ data compilation on synchronised gas ages scales and corrected for offsets between datasets (references are listed in Table 1). The $\delta^{18}$O-H₂O data from the
10   layer-counted NGRIP ice core have been used as an absolute time marker for fast climatic changes (e.g. the Younger Dryas-Holocene transition and the 8.2 ka event) assuming an essentially synchronous change in rapid temperature variations and atmospheric CH₄. The markers at the bottom of the plot indicate the tie points of the synchronised age scales (see Table 3).





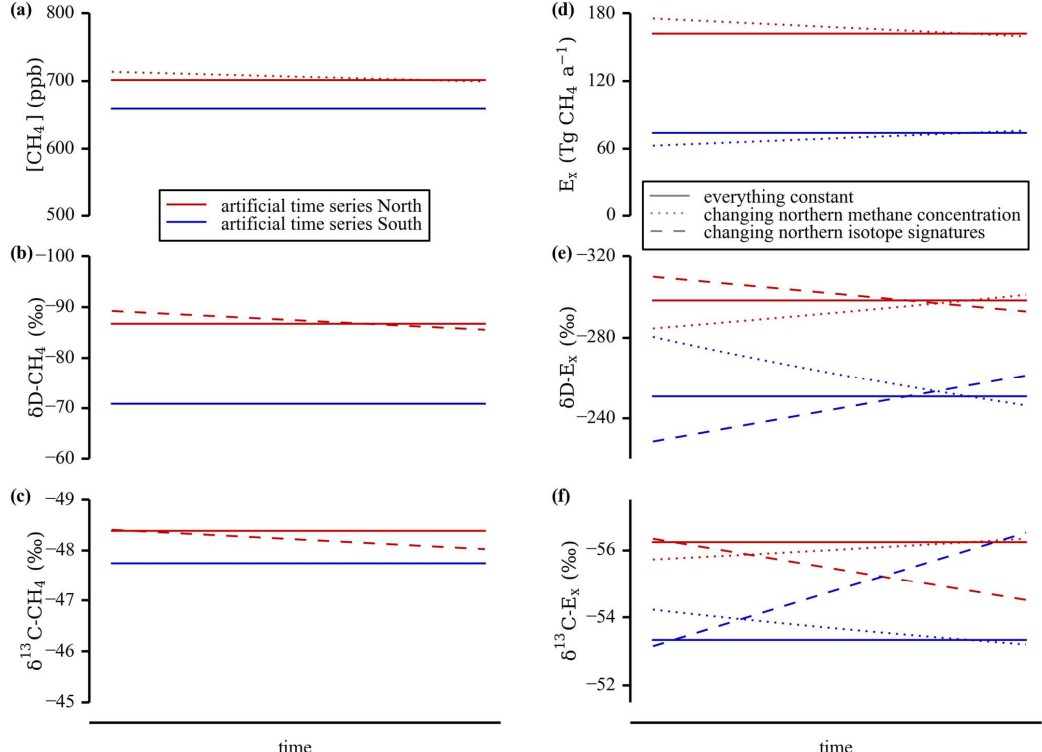

**Figure 2: Artificial time series and deconvolution results.** Artificial tropospheric [CH₄] (a) and stable isotope time series (b), (c) used as input for the deconvolution model. The calculated mean hemispheric CH₄ emission strengths (d) and their isotopic signatures (e), (f) show the dependence on the different input parameters. Three experiments are performed: all input values constant (solid lines), all constant but the northern tropospheric CH₄ isotope values varied (dashed lines) and all constant but [CH₄] varied (dotted lines). To be consistent with the wording in the text, time is running from right to left. Note the inverted y-axes in the panels (b), (c), (e) and (f).





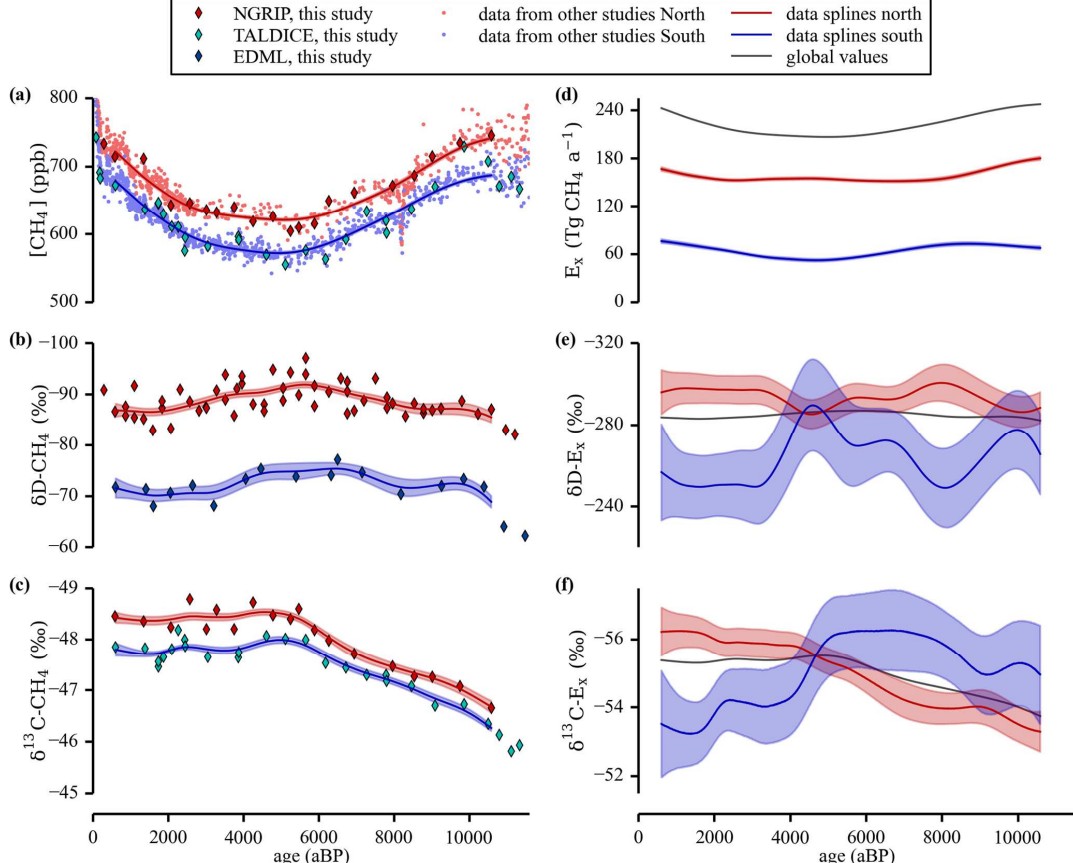

**Figure 3: Data and deconvolution.** The panels on the left (a-c) show the concentration and the two stable isotope signatures of CH$_4$ measured in ice cores representing the tropospheric values over the Holocene. The concentration data are complemented with data from other studies (see Table 1). All three parameters were measured on ice cores from Greenland (red) and form Antarctica (blue and cyan)
5  representing the northern and the southern troposphere, respectively. A spline function with a cutoff period of 3000 years has been used to calculate the smoothed evolution of the tropospheric signals represented by the lines and the $1\sigma$ error bands. A deconvolution model is used to calculate the hemispheric CH$_4$ emissions shown on the panels on the right. In panel (d) the emission strengths are shown for both hemisphere boxes together with the total CH$_4$ emissions. Panel (e) and (f) show the mean isotopic signatures of the emissions for both stable isotopes of CH$_4$. All the error bands represent $1\sigma$ uncertainties. Note the inverted y-axes in the panels (b), (c), (e) and (f).





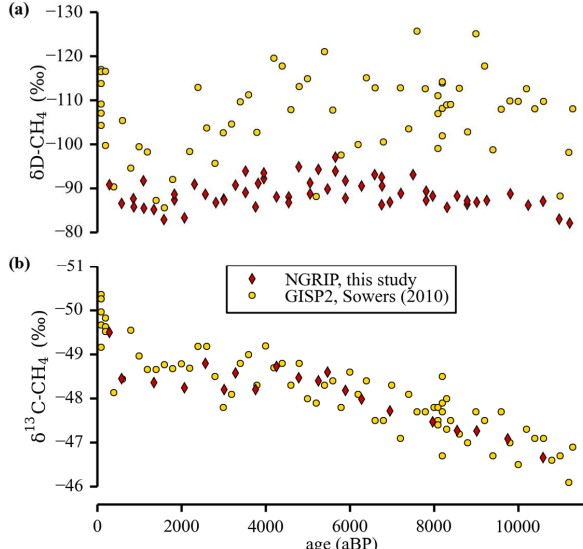

**Figure 4: Comparison of stable CH₄ isotope data.** This study's stable CH₄ isotope values of the NGRIP ice cores are shown together with the data published by Sowers (2010). The $\delta$D-CH₄ data sets (panel (a)) differ in the absolute level and the variance, whereas the $\delta^{13}$C-CH₄ data (panel (b)) are in good agreement. Note that all data are corrected for gravitational fractionation in the firn column. All data of this study are free of Krypton interference, whereas the GISP2 $\delta^{13}$C-CH₄ record has been subsequently corrected for it (Schmitt et al., 2013). Note the inverted y-axes.





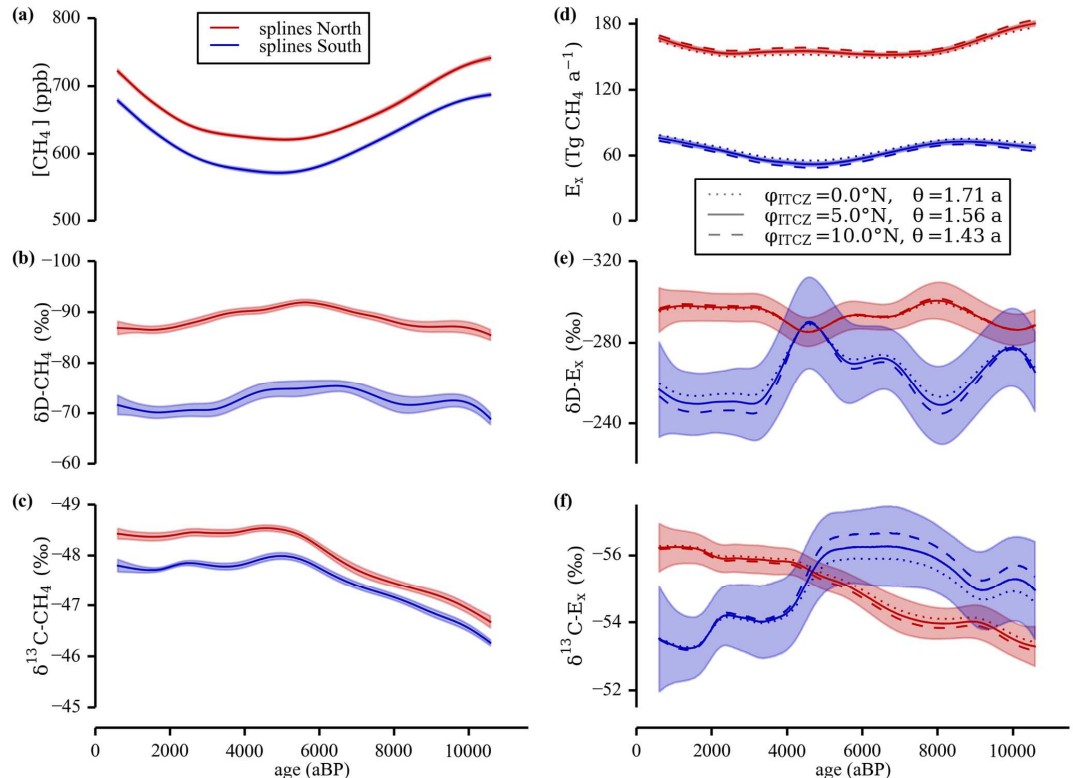

**Figure 5: Sensitivity study "SF₆ calibration".** In the panels (a-c) the data splines with the $1\sigma$ uncertainty bands of the measured tropospheric values are shown. In panels (d-f) the calculated strengths and the isotopic signature of the emissions are shown for the three different $\varphi_{ITCZ}$ and $\theta$ setups to investigate the sensitivity of the deconvolution to the SF₆ calibration. Note the inverted y-axes in the panels (b), (c), (e) and (f).




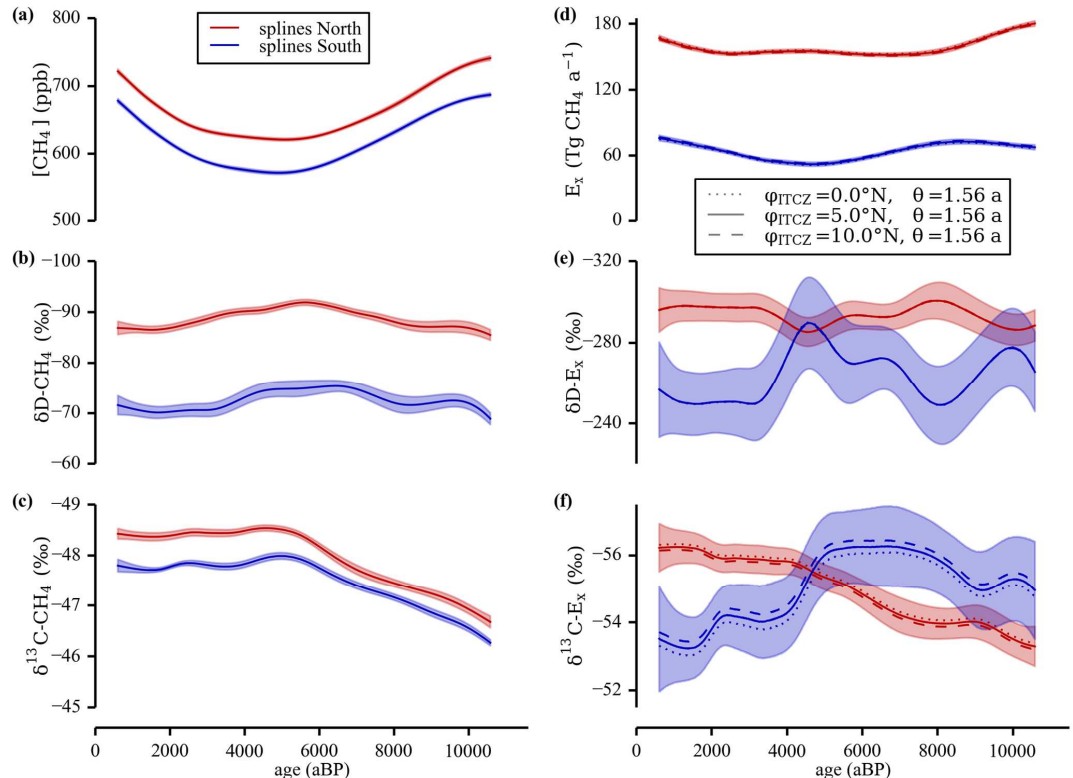

**Figure 6: Sensitivity study "ITCZ position".** In the panels (a-c) the data splines with the $1\sigma$ uncertainty error bands of the measured tropospheric values are shown. In panels (d-f) the calculated strengths and the isotopic signature of the emissions are shown for the three different $\varphi_{ITCZ}$ and the best guess value for $\theta$ to investigate the sensitivity of the deconvolution to a shift of the ITCZ. Note the inverted y-axes in the panels (b), (c), (e) and (f).





**Table 1: Datasets used for CH$_4$ data compilation.** The CH$_4$ offset correction is added to the data to correct for inter-laboratory and time period-related offsets (see section 2.3.2). Additional datasets used for the time matching (e.g. covering the anthropogenic CH$_4$ increase or the Younger Dryas-Holocene transition) are listed. *GRIP data compilation done by J. Flückiger in 2001 containing data from various publications, e.g. Blunier et al. (1995), Chappellaz et al. (1993, 1997).

| Data used for CH$_4$ compilation | | | |
|---|---|---|---|
| Data set name | Data type | Reference | CH$_4$ shift (ppb) |
| NGRIP_Be | NGRIP ice core | This study | 0 |
| GRIP_Fl01 | GRIP ice core | GRIP data Flückiger, 2001* | 17.69 |
| GISP2_Br96 | GISP2 ice core | (Brook et al., 1996) | 34.13 |
| GISP2_Br09 | GISP2 ice core | (Brook, 2009) | −1.81 |
| GISP2_Mi13 | GISP2 ice core | (Mitchell et al., 2013) | −6.15 |
| TALDICE_Be17 | TALDICE ice core | This study | 0 |
| TALDICE_Sc10 | TALDICE ice core | (Schilt et al., 2010) | −2.09 |
| EDML_Sc10 | EDML ice core | (Schilt et al., 2010) | −4.63 |
| WAIS_Mi13a | WAIS (WDC05A) ice core | (Mitchell et al., 2013) | −6.15 |
| WAIS_Mi13b | WAIS (WDC06A) ice core | (Mitchell et al., 2013) | −6.15 |
| EDC_Fl02 | EDC ice core | (Flückiger et al., 2002) | 4.23 |
| | | | |
| Additional data used for time matching | | | |
| Data set name | Data type | Reference | |
| NGRIP_d18O | NGRIP ice core δ$^{18}$O(H$_2$O) | (Rasmussen et al., 2006; Vinther et al., 2006) | |
| NGRIP_Ba12 | NGIRP ice core | (Baumgartner et al., 2014) | |
| NEEM_Bu12 | NEEM firn model | (Buizert et al., 2012) | |
| NOAA_Alert | Alert flasks | (Dlugokencky et al., 2017) | |
| LawDome_Ma06 | Law Dome firn and ice core | (MacFarling Meure et al., 2006) | |
| CapeGrim_csiro | Cape Grim flasks | CSIRO (27.11.2017) | |





Table 2: Ice cores and age scales. Short names of the ice cores and initial gas age scales used for the $CH_4$ compilation.

| Ice core | Initial age scale | References |
|---|---|---|
| NGRIP | AICC2012 ice core chronology | (Bazin et al., 2013; Veres et al., 2013) |
| GRIP | GICC05 | (Rasmussen et al., 2014; Seierstad et al., 2014) |
| GISP2 | Mitchell, Meese/Sowers | (Meese et al., 1994; Mitchell et al., 2013) |
| TALDICE | AICC2012 ice core chronology | (Bazin et al., 2013; Veres et al., 2013) |
| EDML | AICC2012 ice core chronology | (Bazin et al., 2013; Veres et al., 2013) |
| EDC | AICC2012 ice core chronology | (Bazin et al., 2013; Veres et al., 2013) |
| WAIS | Mitchell, Buizert WD2004 | (Buizert et al., 2015; Mitchell et al., 2013) |



**Table 3: Time synchronisation.** Tie points of the reference time scale for the individual ice cores (left column, in aBP) and time corrections (right column, in a) applied with respect to the core dependent time scale listed in Table 2. The correction values are linearly interpolated between the tie points and held constant before the earliest and after the latest tie points.

| NGRIP | | GRIP | | GISP2 | | TALDICE | | EDML | | EDC | | WAIS | |
|---|---|---|---|---|---|---|---|---|---|---|---|---|---|
| 294 | 0 | 273 | −40 | 2752 | −5 | 127 | −134 | 2747 | 0 | 349 | −225 | 2752 | −5 |
| 11800 | −150 | 8157 | 30 | 8172 | −15 | 2835 | −88 | 3754 | 39 | 2959 | −212 | 3753 | 40 |
| | | 11606 | 34 | 11771 | −165 | 3825 | −32 | 4538 | 63 | 3961 | −168 | 4532 | 69 |
| | | | | | | 4596 | 5 | 5279 | 130 | 4741 | −140 | 5269 | 140 |
| | | | | | | 5325 | 84 | 6008 | 215 | 5478 | −69 | 5993 | 230 |
| | | | | | | 6042 | 181 | 6361 | 129 | 6204 | 19 | 6344 | 146 |
| | | | | | | 6389 | 101 | 8107 | 36 | 6555 | −65 | 8079 | 64 |
| | | | | | | 8106 | 37 | 8164 | 22 | 8292 | −149 | 8228 | 28 |
| | | | | | | 8252 | 4 | 8254 | 2 | 8353 | −164 | 9869 | −13 |
| | | | | | | 9876 | −20 | 9875 | −19 | 8438 | −182 | 10391 | 110 |
| | | | | | | 10393 | 108 | 10391 | 110 | 9397 | −179 | 11591 | 49 |
| | | | | | | 11580 | 60 | 11576 | 64 | 9989 | −133 | | |
| | | | | | | | | | | 10462 | 39 | | |
| | | | | | | | | | | 11547 | 93 | | |





**Table 4: GISP2 outliers.** Data points from the two GISP2 data sets by Brook et al. (2009) and Brook (1996) that were removed due to elevated $CH_4$ values probably due to contamination related to the ice sample quality in the brittle zone. Please note that the age and the concentration values are corrected for the age shift and the $CH_4$ correction, respectively.

| Removed data | | | |
|---|---|---|---|
| depth (m) | age (aBP) | $CH_4$ (ppm) | dataset |
| 655.1 | 2731.9 | 698.8 | GISP2_Br09 |
| 736.1 | 3199.1 | 680.8 | GISP2_Br96 |
| 819.7 | 3694.8 | 678.9 | GISP2_Br09 |
| 932.7 | 4411.4 | 661.8 | GISP2_Br09 |
| 943.1 | 4474.0 | 643.8 | GISP2_Br09 |
| 992.3 | 4804.6 | 677.4 | GISP2_Br96 |
| 1019.0 | 4989.5 | 642.0 | GISP2_Br09 |
| 1059.0 | 5272.2 | 631.8 | GISP2_Br09 |
| 1177.3 | 6186.3 | 639.6 | GISP2_Br09 |
| 1194.2 | 6314.4 | 653.8 | GISP2_Br96 |
| 1202.9 | 6382.8 | 652.1 | GISP2_Br96 |
| 1253.9 | 6810.3 | 687.8 | GISP2_Br09 |
| 1274.8 | 6978.6 | 677.8 | GISP2_Br96 |
| 1389.7 | 7993.8 | 675.8 | GISP2_Br96 |
| 1392.0 | 8011.9 | 682.2 | GISP2_Br96 |





**Table 5: Values used in our deconvolution model.** Relative strengths, isotopic fractionations ($^2\varepsilon$ and $^{13}\varepsilon$) and north-south distribution of the individual sink processes (ratio $n/s$) and the parameters for each hemisphere used in our $CH_4$ deconvolution model.

| sink processes | | | | |
|---|---|---|---|---|
| | rel. strength | $^2\varepsilon$ (‰) | $^{13}\varepsilon$ (‰) | ratio $n/s$ |
| $s_{OH}$ | 0.820 | −231 | −5.4 | 246.0/246.0 |
| $s_{strat}$ | 0.092 | −170 | −13.1 | 27.5/27.5 |
| $s_{soil}$ | 0.047 | −80 | −22.0 | 19.6/8.4 |
| $s_{Cl}$ | 0.042 | −315 | −55.0 | 9.6/15.4 |
| | | | | |
| numbers used for deconvolution equations | | | | |
| | $\tau_x(a)$ | $^2\varepsilon_x$ (‰) | $^{13}\varepsilon_x$ (‰) | $r_x$ |
| north box | 7.60 | -218.3 | -7.52 | 0.456 |
| south box | 9.22 | -225.5 | -7.72 | 0.544 |