# Peer review of "Bipolar carbon and hydrogen isotope constraints of the Holocene methane budget"

_Biogeosciences, 2018_

## Referee Comment (RC1) · Anonymous Referee #1 · 1 May 2018

Review of Beck, Biogeosciences, 2018

This paper will be a valuable addition to the literature on the isotopic composition of methane in the past and the Holocene methane cycle. The use of data sets from both hemispheres is very important, as the heuristic modeling in Figure 2 shows. This point could in fact be emphasized a bit more. I do not have any major concerns about the manuscript but do have several smaller issues to bring up that might help improve the clarity of the presentation.

Introduction in general: The introduction is slightly long, and while there is nothing wrong with it some of the sentences and paragraphs could be shortened.

Page 2, line 5: "Low pass" filter may be misinterpreted by those not familiar with ice

cores, I suggest being more specific about what you mean here.

Line 17: Sowers 2010 is probably not the first reference for the bowl shape. Chappellaz et al., 1997?

Line 36: normally elements are not capitalized (deuterium rather than Deuterium).

Line 40: Leaves instead of leave.

Line 42: In instead of on.

Page 3, line 10-14: The sink discussion is a bit cursory particularly since there are papers in the literature about changing sink terms influencing the Holocene budget. For example, Kaplan et al., 2006, GBC. I suggest expanding this section to provide a more detailed defense of keeping the sink strength constant.

Page 4, line 30-31: It is not clear to me what "local average density" means in this sentence. Apparently the local average density is constant but the data are unequally distributed and this sounds contradictory.

Page 5, lines 14-18: It is not clear here if step 2 in the synchronization actually does anything. Is it necessary to have this as a separate component? It reads sort of awkwardly.

Section 2.4 Heading and elsewhere. The term deconvolution is used here, and I think in general this is probably fine, though some would associate it with signal processing, whereas I believe the technique used here would also be called inversion, a term also used in the paper.

Page 7, line 11. A minor point perhaps but is the ppb – burden conversion here consistent with the latest concentration scales?

Equation 3: Although one could calculate rx, out of interest to the reader could you give the value for ITCZ at 5 N to make it clear how much difference it makes?

Page 7, line 5: Is little r defined anywhere near equation 1?

Page 7, line 23: Should these sink terms be constant if the boxes are not the same size?

Page 9, line 6: Indicate if time is left to right or right to left in Figure 2.

Line 23: There may be a better word than "compensate" here, the emissions are required to have a certain time history by the assumptions made.

Line 42-43: This seems repetitive.

Page 11, paragraph 2: The data are interpreted in terms of 2 sigma uncertainty but 1 sigma is plotted. This gets confusing because from the figures there are interesting deviations in the results but the text tells us they are not to be interpreted that way.

Line 16-20: The last two sentences of this paragraph could be reworded. It is first stated that there is a remarkably strong constraint and then stated that the uncertainty does not allow robust conclusions. These two statements seem inconsistent. I think I know what is meant here but it could be made clearer to the reader.

Section 3.1: I may be missing something here. I understand that SF6 in the modern atmosphere can help determine mixing time. I presume that this calculation must by definition choose a value for the mean ITCZ position. One can, I guess, calculate different sets of mixing times for different assumed ITCZ positions given the SF6 data. Is this what what was done for the sensitivity studies? If I have this correct, is this a full exploration of the possibility of changes in atmospheric mixing? Why not just vary the mixing term independently? Is there an assumption here that it only changes because of changes in ITCZ position? What about atmospheric dynamics, could that change?

Figure 3: It is clear enough that red and blue in the right hand panels are for N and S, though this is not actually labeled explicitly in the legend.

---

## Referee Comment (RC2) · Hinrich Schaefer (Referee) · 3 May 2018

Hinrich Schaefer (Referee)

h.schaefer@niwa.co.nz

Review of "Bipolar carbon and hydrogen isotope constraints of the Holocene methane budget" by Jonas Beck et al.

The study presents new measurements of Holocene methane data from ice cores and combines them with literature data to provide novel constraints on the methane budget at different stages during the Holocene. The analysis is based on the combined interpretation of methane mixing ratios, stable carbon and hydrogen isotope ratios, as well as interpolar differences of the three parameters. Interpreted using a 2-box model, the combination of data sets allows to quantify the changing magnitude of hemispheric methane sources as well as their isotopic signatures. This is a highly anticipated ac-

complishment. The findings are important to understand the past methane cycle and are relevant to anticipate future changes. The authors have put a lot of thought and effort into all aspects of the analyses, particularly with regards to poorly known environmental parameters. The manuscript is very well written and structured. This is a very valuable contribution to the field.

I have two points to make. The first is the difference between polar and hemispheric data. The ice core records from Greenland are higher in mixing ratios and more depleted in the rare isotope than the northern hemispheric average, while the reverse is true for Antarctic data and the southern hemisphere. For present day conditions, this can be seen in data products like CarbonTracker-CH4 provided by NOAA-ESRL for mixing ratios and data sets maintained by INSTAAR for delta13C (for an illustration see Figure S1 of Schaefer et al., 2016). As a consequence, the emission estimates from the box model for each hemisphere will be skewed in magnitude and isotopic signature and I would expect the relative changes between different times in the Holocene to be biased as well. It is not straightforward to address this issue because past latitudinal gradients likely differ from modern ones by an unknown margin. However, an attempt should be made, e.g. by correcting the polar data for a probable offset to hemispheric means. For example, Etheridge et al. (1998) and Schaefer et al. (2016) determined the difference between polar and mean global values (for mixing ratio and carbon isotopes, respectively) as a percentage of the interpolar gradient. A similar quantification could be made for hemispheric means.

The other point is a suggestion to derive more information from the presented data. The isotopic signature of the emissions that have changed between two periods can be derived by mass balance according to E(total)*D(total)=E(initial)*D(initial) + E(change)*D(change); where E and D stand for the magnitude and isotopic signature of emissions, respectively. This would enable a more robust discussion of the changes, e.g. by comparing the isotope signatures to those of major emission types. Statements like the one on page 13, lines 37-38, can then be based on specific numbers and discussed in the context of known source isotopic signatures.

Minor comments are listed in the following.

Page 1, line 36 (1/36): these numbers don't match with Table 1 of Naik et al. (2013). Arguably, observation-based estimates like the one from Prather et al. (2012) are more reliable than model estimates.

1/37: even for current emission estimates there is a sizeable difference between bottom-up and top-down estimates (Kirschke et al., 2013; Saunois et al., 2017).

3/10-14: The sink changes between modern day and pre-industrial times as studied by Naik et al. (2013) are more relevant here (although the author's point remains valid that sink changes are unlikely to have significant impact). It cannot be ruled out that the ratio of sinks changed throughout the study period, e.g., due to different sensitivity to changing $CH_4$ mixing ratio or environmental conditions. Such changes are likely to be minor or negligible, but the point should be made.

Table 1: it is interesting that different time series from the same lab can have different offsets to the spline. This means that the offset results are not transferable to future studies and raises questions as to the cause of the offsets. It may or may not be worth pointing that out.

Section 2.3.3: the resulting GISP2 data sets are likely of interest for future studies and I wonder if they can be made available. However, that may be problematic because the credit would have to be shared between Ed Brook as supplier of the original data and the current authors for their data analysis. Just a thought.

Section 2.3.: there should be an additional subsection to derive hemispheric averages from the polar data, as discussed above.

7/18 and following lines: the parameterization of sinks and associated fractionation is complex and without definitive solution. I see no problems with the particular choice of values by the authors, but the following points should be made clear. (i) Other

studies arrive at or use different numbers for individual parameters that are equally as valid (e.g., preindustrial tau=10.1±1.7 a, Naik et al., 2013; tropospheric expression of stratospheric fractionation = -3‰ Lassey et al., 2000). (ii) It is uncertain if modern estimates can be used for Holocene conditions. (iii) It cannot be ruled out that changes in environmental conditions and atmospheric chemistry – including $CH_4$ mixing ratios – changed the sink and its total fractionation throughout the studied period. (iv) The exact choice of sink parameters is of minor importance as long as the focus of the interpretation are relative changes in emissions.

7/26-27: please provide a specific reference for the OH fractionation coefficients, i.e., Saueressig et al., (2001)

Section 2.4.1.: if the authors correct the input data to hemispheric averages, then the transport times derived from SF6 must be adjusted.

9/10-12: this value is likely to change when the model is run for hemispheric averages rather than high-latitude data.

9/15-17: I don't follow the description of this experiment. Is the linear change between minimum IPD at start of the run to maximum IPD at its end (or vice-versa)? Please clarify.

11/1-6: please revise these results for hemispheric averages instead of polar data.

13/37-39: it is not possible to evaluate the statement unless the authors provide quantification. This can be done by calculating the isotopic signatures of the additional emissions by isotope mass balance (as mentioned above).

13/38-41: tropical wetland emissions are more 13C-rich than other biogenic sources, so qualitatively the trend can be explained by their greater prominence. To evaluate whether an additional 13C-rich source is increasing requires the mass balance results. If a simultaneous rise of wetland and fire emissions is invoked, then the discussion should explain how increasing precipitation that enhances wetland $CH_4$ production

also leads to more fires, which occur in dry conditions. Greater fuel supply? More smoldering fires?

14/11-13: does this statement (isotopic changes are due to changing source signatures) stand for all time periods? That would contrast with the discussion in Section 4.3.

Fig. 3: the IPDs are a crucial parameter derived in this study. I recommend showing them in the graph (or a separate graph).

Fig. 3d: the temporal changes in the three emission lines are hard to see on the presented scale, making it difficult to follow the discussion in Sections 4.1.-4.3. Consider providing an additional or alternative graph with more detail on changes in emission magnitude.

Table 5: please provide references for the various literature values. The listed value for epsilon(OH) is from an older study (Cantrell et al., 1994), a more precise estimate has been published by Saueressig et al. (2001).

Minor corrections:

5/34 "can complicate" not ("can complicates").

6/14: should this read "residual" rather than "residuum"?

8/4: better: "in both time and magnitude"

9/41: "with a full decoupling"

10/31-32: consider rewording, e.g.: "at about the same time the d13C-CH4 signals undergo a step change from . . ."

14/8-9: "This shows the value. . ." consider rewording this sentence.

14/19: ". . .without such evidence." Consider rewording.

Fig. 2: consider arrows for the time axes.

[Figure]

---

## Author Comment (AC1) · 15 Jun 2018

We thank the reviewer for this constructive and detailed review. Below we respond to each point in blue text and propose altered text in italics.

This paper will be a valuable addition to the literature on the isotopic composition of methane in the past and the Holocene methane cycle. The use of data sets from both hemispheres is very important, as the heuristic modeling in Figure 2 shows. This point could in fact be emphasized a bit more. I do not have any major concerns about the manuscript but do have several smaller issues to bring up that might help improve the clarity of the presentation.

Introduction in general: The introduction is slightly long, and while there is nothing wrong with it some of the sentences and paragraphs could be shortened.

We went through the introduction carefully and besides some minor changes did not change much. We think that all the information given is required to put the study in perspective and therefore we are not able to shorten it considerably.

Page 2, line 5: "Low pass" filter may be misinterpreted by those not familiar with ice cores, I suggest being more specific about what you mean here.

We will change the text to be more specific:
*Only polar ice cores can provide this information as they represent a direct, albeit through the bubble enclosure process in the firn low-pass filtered archive of trapped air and, therefore, offer the possibility to investigate the atmospheric composition of the past.*

Line 17: Sowers 2010 is probably not the first reference for the bowl shape. Chappellaz et al., 1997?

We will change the citation to *Blunier et al., 1995* instead of Sowers 2010

Line 36: normally elements are not capitalized (deuterium rather than Deuterium).

Yes, we will change that.

Line 40: Leaves instead of leave.

Yes, we will change that.

Line 42: In instead of on.

Yes, we will change that.

Page 3, line 10-14: The sink discussion is a bit cursory particularly since there are papers in the literature about changing sink terms influencing the Holocene budget. For example, Kaplan et al., 2006, GBC. I suggest expanding this section to provide a more detailed defense of keeping the sink strength constant.

We will change this paragraph:
*As mentioned above, changes in the sinks leave an imprint on the isotopic composition of CH4 in the atmosphere as well. However, while minor changes in the relative importance of the sink processes cannot be ruled out, in general, atmospheric chemistry models point to small changes in total lifetime (Kaplan et al., 2006; Levine et al., 2011a) and we also assume the relative contribution of individual sinks to be rather constant over the Holocene when relatively stable climate conditions prevailed. Therefore, the discussion of this study centres around CH4 source processes, hence emissions of different CH4 source categories.*

Page 4, line 30-31: It is not clear to me what "local average density" means in this sentence. Apparently the local average density is constant but the data are unequally distributed and this sounds contradictory.

We will change the text as follows:
*The down-sampled data are still unequally distributed, but on average the data density is similar over the whole time period studied.*

Page 5, lines 14-18: It is not clear here if step 2 in the synchronization actually does anything. Is it necessary to have this as a separate component? It reads sort of awkwardly.

Yes, we did not find any additional tie points when we tried to align the different datasets within the two hemispheres. However, we think it is important to mention that we could not find any incongruities with the given data resolutions.

Section 2.4 Heading and elsewhere. The term deconvolution is used here, and I think in general this is probably fine, though some would associate it with signal processing, whereas I believe the technique used here would also be called inversion, a term also used in the paper.

We will change the term deconvolution to inversion throughout the manuscript.

Page 7, line 11. A minor point perhaps but is the ppb – burden conversion here consistent with the latest concentration scales?

The ppb-burden conversion is the same as in the 4 box model described in Fischer et al., 2008 and in Baumgartner et al., 2012. This allows a direct comparison between the different model results. By using more recent estimates of the atmospheric $CH_4$ burden and the mean $CH_4$ concentration we end up with a slightly different conversion factor (e.g. using numbers by Dalsøren et al., 2016 we get $m^* = 2.6 \ to \ 2.7 \ Tg/ppb$ instead of $m^* = 2.9 \ Tg/ppb$). However, since the focus of this study is on relative changes and not on absolute values this does not affect our conclusions.
We will change the text as follows:
*The parameter m^\*=m_0/c_0 translates atmospheric concentrations (ppb) to total atmospheric inventories (in Tg) using the mean atmospheric concentration value c_0=1650 ppb of the year 1987. Here we use the corresponding global CH4 burden m_0=4800 Tg by Steele et al. (1992) that was also used in previous ice core studies (Baumgartner et al., 2012; Fischer et al., 2008). Note that this value is 7-10% higher than more recent estimates (Dalsøren et al., 2016) and accordingly the derived absolute emissions may be 7-10% overestimated. However, we focus our interpretation on relative emission changes, which are not affected by this scaling factor.*

Equation 3: Although one could calculate rx, out of interest to the reader could you give the value for ITCZ at 5 N to make it clear how much difference it makes?

OK, we will include *(r_n=0.46, r_s=0.54 for φ_ITCZ=5°N).*

Page 7, line 5: Is little r defined anywhere near equation 1?

The ratio of the box volume r_x is defined in Eq. 3.

Page 7, line 23: Should these sink terms be constant if the boxes are not the same size?

The sink processes are quantified by the fluxes s_OH, s_strat, s_soil and s_Cl (in Tg $CH_4$ per year). According to where the different processes are thought to take place they are distributed over the two hemispheres. The size of the hemispheres (r_x) becomes important when the hemispheric lifetime (and sink fractionations) are calculated (Eq. 4). We kept the box size hence also the relative sink contributions constant in time in our standard run.

Page 9, line 6: Indicate if time is left to right or right to left in Figure 2.

By omitting the direction of the time flow we tried to indicate that the artificial time series could principally be interpreted in both directions. But we will add arrows indicating that time is running from right to left (to be consistent with all other figures) to avoid confusion.

Line 23: There may be a better word than "compensate" here, the emissions are required to have a certain time history by the assumptions made.

We will change the text to:
*At the same time the emissions in the isotopically lighter CH4 mixed from the northern into the southern box requires the southern box emissions to become more enriched in the heavy isotope (isotopically "heavier", equivalent to higher δ-values) over time.*

Line 42-43: This seems repetitive.

Yes, maybe this is slightly repetitive. But we think it is important to point it out clearly that although the chosen ranges are based on the Holocene data, the explained changes are not realistic.

Page 11, paragraph 2: The data are interpreted in terms of 2 sigma uncertainty but 1 sigma is plotted. This gets confusing because from the figures there are interesting deviations in the results but the text tells us they are not to be interpreted that way.

Yes, we will add 2 sigma uncertainty ranges in Figure 3.

Line 16-20: The last two sentences of this paragraph could be reworded. It is first stated that there is a remarkably strong constraint and then stated that the uncertainty does not allow robust conclusions. These two statements seem inconsistent. I think I know what is meant here but it could be made clearer to the reader.

We will change the wording in this paragraph to clarify:
*The absence of a long-term trend in δD-Ex over the Holocene with the significant changes in the atmospheric [CH4] is remarkable and a strong constraint on the average Holocene CH4 budget. However, the large uncertainty in δD-Ex does not allow us to make robust conclusions about millennial variations in the hydrogen isotopic signature of CH4 emissions.*

Section 3.1: I may be missing something here. I understand that SF6 in the modern atmosphere can help determine mixing time. I presume that this calculation must by definition choose a value for the mean ITCZ position. One can, I guess, calculate different sets of mixing times for different assumed ITCZ positions given the SF6 data. Is this what what was done for the sensitivity studies? If I have this correct, is this a full exploration of the possibility of changes in atmospheric mixing? Why not just vary the mixing term independently? Is there an assumption here that it only changes because of changes in ITCZ position? What about atmospheric dynamics, could that change?

Yes, our SF6 calibration approach is based on modern values and thus represents todays atmospheric dynamics. Thus the mixing time derived from the SF6 calibration is a modern value and we keep this constant over time as the (tropical) climate conditions are rather stable over the Holocene except for slight shifts in the ITCZ position.

Figure 3: It is clear enough that red and blue in the right hand panels are for N and S, though this is not actually labeled explicitly in the legend.

In the figure caption we will add:
*Red colours refer to northern hemisphere records, blue colours to southern hemisphere records throughout the manuscript.*

---

## Author Comment (AC2) · 15 Jun 2018

We thank Hinrich Schaefer for this constructive and detailed review. Below we respond to each point in blue text and propose altered text in italics.

The study presents new measurements of Holocene methane data from ice cores and combines them with literature data to provide novel constraints on the methane budget at different stages during the Holocene. The analysis is based on the combined interpretation of methane mixing ratios, stable carbon and hydrogen isotope ratios, as well as interpolar differences of the three parameters. Interpreted using a 2-box model, the combination of data sets allows to quantify the changing magnitude of hemispheric methane sources as well as their isotopic signatures. This is a highly anticipated accomplishment. The findings are important to understand the past methane cycle and are relevant to anticipate future changes. The authors have put a lot of thought and effort into all aspects of the analyses, particularly with regards to poorly known environmental parameters. The manuscript is very well written and structured. This is a very valuable contribution to the field.

I have two points to make. The first is the difference between polar and hemispheric data. The ice core records from Greenland are higher in mixing ratios and more depleted in the rare isotope than the northern hemispheric average, while the reverse is true for Antarctic data and the southern hemisphere. For present day conditions, this can be seen in data products like CarbonTracker-CH4 provided by NOAA-ESRL for mixing ratios and data sets maintained by INSTAAR for delta13C (for an illustration see Figure S1 of Schaefer et al., 2016). As a consequence, the emission estimates from the box model for each hemisphere will be skewed in magnitude and isotopic signature and I would expect the relative changes between different times in the Holocene to be biased as well. It is not straightforward to address this issue because past latitudinal gradients likely differ from modern ones by an unknown margin. However, an attempt should be made, e.g. by correcting the polar data for a probable offset to hemispheric means. For example, Etheridge et al. (1998) and Schaefer et al. (2016) determined the difference between polar and mean global values (for mixing ratio and carbon isotopes, respectively) as a percentage of the interpolar gradient. A similar quantification could be made for hemispheric means.

Yes, the values measured on polar ice cores do not represent mean hemispheric signals. And we also agree that the recommended corrections describe one possible way to address the issue. For this study, however, we decided to adapt the atmospheric transport in our deconvolution model to already account for this effect. By using high altitude (polar) SF6 data for the calibration of the polar exchange time $\theta$ which is then used to quantify the CH4 exchange between the boxes we inherently correct for the described effect of using polar instead of mean hemispheric values. As a result, with 1.56 years the inter polar exchange time is larger than the literature values for the interhemispheric exchange time of 1.3 years (Geller et al., 1997).

In response to the review comment we tried to test the strategy recommended by Hinrich Schaefer and to use the knowledge about the modern CH4 distribution and the past interpolar CH4 difference to scale the measured values. Using the CH4 concentration data from the numerous measurement stations we calculated the latitudinal CH4 concentration distribution over the years 1997-2017 using the monthly mean data from ftp://aftp.cmdl.noaa.gov/data/trace_gases/ch4/flask/surface/ (Dlugokencky et al., 2017). Based on the latitude weighted mean hemispheric CH4 concentrations (with $\varphi_{ITCZ} = 5°N$) and the polar CH4 concentrations we calculated the average correction factor for both hemisphere boxes according to the relationship

$c_{hemisphere} = c_{polar} + \omega * IPD$ and got: $\omega_n = -1.77, \omega_s = 0.71$

With the measured polar CH4 concentration for both hemispheres (and thus the IPD) this allows us to translate the polar CH4 data to hemispheric mean values. Assuming the same relative latitudinal distribution for δD-CH4 and δ13C-CH4 the correction can be also applied to the stable isotope time series.

However, to be consistent we also have to adapt the exchange time $\theta$ since it should now represent the

exchange time between the two hemispheres. We tried to do so by following the same SF6 calibration explained in the manuscript using mean hemispheric SF6 concentrations based on SF6 data by Levin et al. (2010). By doing so, we end up with a hemispheric exchange time of only $\theta = 1.02 \pm 0.06$ years. This value is not in line with the literature. The low value of the exchange time is most likely the result of the insufficient latitudinal resolution of the SF6 data, in particular in the low latitudes. As a consequence the calculated mean hemispheric SF6 levels of the two boxes are shifted towards each other (underestimated $IPD_{SF6}$) leading to the overestimation of the hemispheric air mass exchange. In summary, while the suggestion by the reviewer represents an alternative strategy, we cannot reliably calibrate our model using SF6 in this case.

Due to this difficulties we decided to stick to our method to keep the ice core time series and use the interpolar exchange time for the deconvolution. However, the corresponding sections in the manuscript are adapted to point out this issue and our solution more explicitly. Note, that we will use the expression "interpolar exchange time" throughout the manuscript to be consistent with our calibration approach.

The other point is a suggestion to derive more information from the presented data. The isotopic signature of the emissions that have changed between two periods can be derived by mass balance according to E(total)*D(total)=E(initial)*D(initial) + E(change)*D(change); where E and D stand for the magnitude and isotopic signature of emissions, respectively. This would enable a more robust discussion of the changes, e.g. by comparing the isotope signatures to those of major emission types. Statements like the one on page 13, lines 37-38, can then be based on specific numbers and discussed in the context of known source isotopic signatures.

So far we consciously refrained from such quantitative analysis of the inversion values due to the following reasons: (1) As we describe throughout the manuscript, the box model is based on assumptions which only allow robust conclusions on relative changes. (2) The mass balance approach proposed by the reviewer assumes constant isotopic signatures of the initial $CH_4$ emissions, which cannot be guaranteed. Nonetheless, we decided to make the calculation as proposed and discuss the caveats as follows:
*The observed shift in δD-Es and especially δ13C-Es towards heavier values in this interval cannot be explained simply by increased tropical wetland emissions in the Amazon region but requires also an enhancement of a southern hemisphere CH4 source strongly enriched in 13C. Using a simple mass balance approach (E_total·iδ_total= E_initial·iδ_initial +E_additional·iδ_additional) allows to determine a δ13C signature of the additional CH4 emissions of 19 Tg CH4/a to be about −45.5 ‰ which is isotopically heavier than tropical wetland CH4 emissions (−56.8 ‰ according to Whiticar and Schaefer (2007)). Note, however, that this quantitative approach is based on the assumption of unchanged isotope signatures of the initial CH4 emissions and therefore needs to be taken with care.*

Minor comments are listed in the following.

Page 1, line 36 (1/36): these numbers don't match with Table 1 of Naik et al. (2013). Arguably, observation-based estimates like the one from Prather et al. (2012) are more reliable than model estimates.

We will change the number and the citation:
*… which is 9.1 ± 0.9 years (Prather et al., 2012).*

1/37: even for current emission estimates there is a sizeable difference between bottom-up and top-down estimates (Kirschke et al., 2013; Saunois et al., 2017).

Yes, and that is what this sentence also implies. We will clarify:
*Despite the fact that the total CH4 emissions today are well known, there is still considerable debate about the individual contributions of (in particular small) CH4 source and their variability, e.g. evidenced by the mismatch of bottom-up and top-down estimates of the total CH4 emissions (Crill and Thornton, 2017; Kirschke et al., 2013; Saunois et al., 2017).*

3/10-14: The sink changes between modern day and pre-industrial times as studied by Naik et al. (2013) are more relevant here (although the author's point remains valid that sink changes are unlikely to have significant impact). It cannot be ruled out that the ratio of sinks changed throughout the study period, e.g., due to different sensitivity to changing CH4 mixing ratio or environmental conditions. Such changes are likely to be minor or negligible, but the point should be made.

We will change this paragraph as follows:
*As mentioned above, changes in the sinks leave an imprint on the isotopic composition of CH4 in the atmosphere. However, while minor changes in the relative importance of the sink processes cannot be ruled out, in general, atmospheric chemistry models point to small changes in total lifetime (Kaplan et al., 2006; Levine et al., 2011a) and we also assume the relative contribution of individual sinks to be rather constant over the Holocene when relatively stable climate conditions prevailed. Therefore, the discussion of this study centres around CH4 source processes, hence emissions of different CH4 source categories.*

Table 1: it is interesting that different time series from the same lab can have different offsets to the spline. This means that the offset results are not transferable to future studies and raises questions as to the cause of the offsets. It may or may not be worth pointing that out.

Yes, the offsets are not to be transferred to any other datasets. Therefore, we tried to highlight the value of interpolar data sets measured not only with the same measurement setup, but also during the same measurement campaign. The text in the conclusion section (14/32) will be adapted as follows:
*Better IPD_CH4 data measured on the same analytical system during the same measurement campaign in high temporal resolution, which ensures an accurate synchronisation (such as the data by Mitchell et al. (2013) for the last 2,800 years), would bring us a large step forward in further constraining the Holocene CH4 cycle. This is especially true for the Greenland ice cores within the brittle ice zone, e.g. GISP2 records for depths between 650 and 1400 m, equivalent to the time window 2.7-8.1 ka BP where individual samples subject to modern air intrusion may compromise the Greenland CH4 record. The new EGRIP ice core currently drilled in northeast Greenland with much lower accumulation, hence a brittle ice zone located at older ages may help to improve the CH4 record in this time interval in the future.*

Section 2.3.3: the resulting GISP2 data sets are likely of interest for future studies and I wonder if they can be made available. However, that may be problematic because the credit would have to be shared between Ed Brook as supplier of the original data and the current authors for their data analysis. Just a thought.

The GISP2 data are published and available at http://nsidc.org/data/nsidc-0440. We use and cite the data in the same manner as other data used for our CH$_4$ data compilation.

Section 2.3.: there should be an additional subsection to derive hemispheric averages from the polar data, as discussed above.

See comment above.

7/18 and following lines: the parameterization of sinks and associated fractionation is complex and without definitive solution. I see no problems with the particular choice of values by the authors, but the following points should be made clear. (i) Other studies arrive at or use different numbers for ndividual parameters that are equally as valid (e.g., preindustrial tau=10.1±1.7 a, Naik et al., 2013; tropospheric expression of stratospheric fractionation = -3‰ Lassey et al., 2000). (ii) It is uncertain if modern estimates can be used for Holocene conditions. (iii) It cannot be ruled out that changes in environmental conditions and atmospheric chemistry – including CH4 mixing ratios – changed the sink and its total fractionation throughout the studied period. (iv) The exact choice of sink parameters is of minor importance as long as the focus of the interpretation are relative changes in emissions.

Regarding the choice of the lifetime we will add the following statement (7/19):
*Again, using other estimates of the atmospheric lifetime (Naik et al., 2013; Prather et al., 2012) changes the absolute emissions estimates but does not affect our conclusions on relative emission changes.*
Further we will clarify:
*The chosen values for the atmospheric CH4 lifetime and the strengths and the distribution of the sinks are just a best guess based on the cited literature. Hence, other studies end up with slightly different values. Note also that we assume the model parameters to remain temporally unchanged over the Holocene.*

7/26-27: please provide a specific reference for the OH fractionation coefficients, i.e., Saueressig et al., (2001)

Yes, we will add Saueressig et al. (2001) to the references.

Section 2.4.1.: if the authors correct the input data to hemispheric averages, then the transport times derived from SF6 must be adjusted.

See comment above.

9/10-12: this value is likely to change when the model is run for hemispheric averages rather than high-latitude data.

See comment above.

9/15-17: I don't follow the description of this experiment. Is the linear change between minimum IPD at start of the run to maximum IPD at its end (or vice-versa)? Please clarify.

We will clarify to:
*For the second experiment (dashed lines in Figure 2), the IPD of the tropospheric isotope signatures (IPD_δD and IPD_δ13C) have been changed while the concentrations remained unchanged. By only altering the northern stable isotope signal the isotopic IPD's are changing from the minimal to the maximal observed IPD_δD and IPDδ_13C values in the splined isotope data, respectively (dashed lines in (b) and (c)).*

11/1-6: please revise these results for hemispheric averages instead of polar data.

See comment above.

13/37-39: it is not possible to evaluate the statement unless the authors provide quantification. This can be done by calculating the isotopic signatures of the additional emissions by isotope mass balance (as mentioned above).

See comment above.

13/38-41: tropical wetland emissions are more 13C-rich than other biogenic sources, so qualitatively the trend can be explained by their greater prominence. To evaluate whether an additional 13C-rich source is increasing requires the mass balance results. If a simultaneous rise of wetland and fire emissions is invoked, then the discussion should explain how increasing precipitation that enhances wetland CH4 production also leads to more fires, which occur in dry conditions. Greater fuel supply? More smoldering fires?

We will complement as follows:
*Increasing emissions from tropical and subtropical wild fires in the southern hemisphere, which are also documented in charcoal records (Daniau et al., 2012), could have become increasingly important, e.g. due to enhanced fuel production or higher CH4 emission efficiency (more smoldering fires) both caused by wetter climatic conditions, and would readily explain the joint information*

*derived from inversion results on southern hemisphere CH4 emissions and its stable isotopic signatures.*

14/11-13: does this statement (isotopic changes are due to changing source signatures) stand for all time periods? That would contrast with the discussion in Section 4.3.

We will clarify to:
*Therefore, we attribute the observed long-term changes of the atmospheric CH4 isotopes over the first half of the Holocene to shifts in the isotopic source signatures of individual CH4 sources rather than to changes in the global CH4 source mix.*

Fig. 3: the IPDs are a crucial parameter derived in this study. I recommend showing them in the graph (or a separate graph).

We will add panels showing the IPD's in figure 3.

Fig. 3d: the temporal changes in the three emission lines are hard to see on the presented scale, making it difficult to follow the discussion in Sections 4.1.-4.3. Consider providing an additional or alternative graph with more detail on changes in emission magnitude.

OK, we will increase the height of figure 3 (also to get enough space for the additional IPD panels) which allows us also to stretch the axes of the inversion panels. Additionally, we will add grid lines in panel (d) to better visualise the emissions changes.

Table 5: please provide references for the various literature values. The listed value for epsilon(OH) is from an older study (Cantrell et al., 1994), a more precise estimate has been published by Saueressig et al. (2001).

We will add the following citations in the caption of table 5:
*(Brenninkmeijer et al., 1995; Cantrell et al., 1990; Feilberg et al., 2005; Gierczak et al.,1997; Irion et al., 1996; Tyler et al., 1994; Quay et al., 1999)*
We are aware that numbers from more recent studies (e.g. Saueressig et al. (2001)) exist. The values we use have also been used for the 4 box model by Fischer et al. (2008) and Bock et al. (2017). For comparison reason we decided to not to change them.

Minor corrections:

5/34 "can complicate" not ("can complicates").

Yes, we will change that.

6/14: should this read "residual" rather than "residuum"?

Yes, we will change that.

8/4: better: "in both time and magnitude"

Yes, we will change that.

9/41: "with a full decoupling"

Yes, we will change that.

10/31-32: consider rewording, e.g.: "at about the same time the d13C-CH4 signals undergo a step change from..."

We will change the text as follows:
*At about the same time the IPD_$\delta13C$ changes quickly from $-0.35$ ‰ to $-0.63$ ‰, which is rather constant before and after the point of inflection.*

14/8-9: "This shows the value..." consider rewording this sentence.

We will change the text as follows:
*This shows the value of our multi isotope approach in order to avoid drawing the wrong conclusions.*

14/19: "...without such evidence." Consider rewording.

We will change the text as follows:
*An alternative explanation calling upon a significant decrease of CH4 emissions from thermokarst lakes would require compensation by other processes to fulfil the δ13C constraint, e.g. a substantial decline of wild fire activity in the northern hemisphere.*

Fig. 2: consider arrows for the time axes.

Principally the figure has no orientation since the shown data fulfil the deconvolution equation in both directions (time running from left to right and vice-versa). To avoid unnecessary confusion, we will apply an arrow showing the direction of the time (in the same direction as it is the case for the other figures).

---

## Author Response (AR1)

Dear Kirsten Thonicke,

We thank you for the comments. Below we respond to your points in blue and propose corresponding text in the manuscript in italics.

Please mention this point in the text of the revised manuscript:
Reviewer 1: Page 5, lines 14-18: It is not clear here if step 2 in the synchronization actually does anything. Is it necessary to have this as a separate component? It reads sort of awkwardly.
Your response. Yes, we did not find any additional tie points when we tried to align the different datasets within the two hemispheres. However, we think it is important to mention that we could not find any incongruities with the given data resolutions.

We slightly changed passage in the manuscript which now reads as follows:
*However, within both hemispheres the temporal alignments of all the distinguishable $CH_4$ features were already within the limitation of the data resolution. Therefore, no additional tie points could be defined at this stage, within each hemisphere.*

Please check if you can include your response to the first major point raised by Hinrich Schaefer in his review into the discussion.

After discussing our model calibration using $SF_6$ data to calculate the polar mixing time (section 2.4.1) we added the following paragraph:
*Another valid strategy to cope with the ice core data representing polar tropospheric values would be to correct the data to represent mean hemispheric ($\varphi\_ITCZ=5.0°N$) values (similarly as it has been done by Etheridge et al. (1998) and by Schaefer et al. (2016) to derive a global average signal from ice core $CH_4$ and $\delta^{13}C$-$CH_4$ data) based on the knowledge about the modern $CH_4$ concentration distribution (e.g. as provided by Dlugokencky et al. (2017)). However, using the corrected data for the box-model inversion also requires the corresponding calculation of $\theta$ (to represent the hemispheric mixing time) using mean hemispheric $SF_6$ concentrations. The lower spatial resolution of the $SF_6$ concentration data (especially in the low latitudes) biases the mean hemispheric $SF_6$ values and thus does not allow a satisfying calculation of a hemispheric exchange time. Therefore, we decided not to follow this approach and use polar values in combination with the polar exchange time (as explained above) for our inversion study.*

Best wishes,
Jonas Beck

**Bipolar carbon and hydrogen isotope constraints of the Holocene methane budget**

Jonas Beck[1,2], Michael Bock[1,2], Jochen Schmitt[1,2], Barbara Seth[1,2], Thomas Blunier[3], Hubertus Fischer[1,2]

[1]Climate and Environmental Physics, Physics Institute, University of Bern, Sidlerstrasse 5, 3012 Bern, Switzerland
[2]Oeschger Centre for Climate Change Research, University of Bern, 3012 Bern, Switzerland
[3]Centre for Ice and Climate, Niels Bohr Institute, University of Copenhagen, Juliane Maries Vej 30, 2100 Copenhagen, Denmark

*Correspondence to*: Jonas Beck (beck@climate.unibe.ch)

**Abstract.** Atmospheric methane concentration shows a well-known decrease over the first half of the Holocene following the northern hemisphere summer insolation before it started to increase again to preindustrial values. There is a debate about what caused this change in the methane concentration trend, in particular, whether an early anthropogenic influence or natural emissions led to the reversal of the atmospheric $CH_4$ concentration. Here, we present new methane concentration and stable hydrogen and carbon isotope data measured on ice core samples from both Greenland and Antarctica over the Holocene. With the help of a two-box model and the full suite of $CH_4$ parameters, the new data allow us to quantify the total methane emissions in the northern and southern hemispheres separately as well as their isotopic signatures, while interpretation of isotopic records of only one hemisphere may lead to erroneous conclusions. For the first half of the Holocene our results indicate a̶a̶n̶ asynchronous decrease in northern and southern hemisphere $CH_4$ emissions by more than 30 Tg $CH_4$/y̶r̶a in total accompanied by a drop in the northern carbon isotopic source signature of about −3 ‰. This cannot be explained by a change in the source mix alone, but requires shifts in the isotopic signature of the sources themselves caused by changes in the precursor material for the methane production. In the second half of the Holocene global $CH_4$ emissions increased by about 30 Tg $CH_4$/y̶r̶a, while preindustrial isotopic emission signatures remained more a less constant. However, our results show that t̶h̶e̶this early increase of methane emissions ̶s̶t̶a̶r̶t̶i̶n̶g̶ ̶i̶n̶ ̶t̶h̶e̶ ̶m̶i̶d̶ ̶H̶o̶l̶o̶c̶e̶n̶e̶ took place in the southern hemisphere, while northern hemisphere emissions started to increase only about 2000 years ago. Accordingly, natural emissions in the southern tropics appear to be the main cause of the $CH_4$ increase starting 5000 years ̶a̶g̶o̶ ̶i̶n̶ ̶c̶o̶n̶t̶r̶a̶d̶i̶c̶t̶i̶o̶n̶ ̶t̶o̶, not supporting an early anthropogenic influence on the global methane budget by East Asian land use changes.

**1 Introduction**

Atmospheric methane ($CH_4$) is a potent greenhouse gas and its concentrations are strongly coupled to the Earth's climate system. Due to the human influence on the Earth system, the $CH_4$ concentration ([$CH_4$]) in the atmosphere has increased by a factor of 2.5 (relative to the preindustrial level) over the last centuries and contributes significantly to the human-induced radiative forcing (Dlugokencky et al., 2005; Etheridge et al., 1998). Today both anthropogenic $CH_4$ sources (rice agriculture, livestock, fossil fuel production, anthropogenic biomass burning and landfills) and natural $CH_4$ sources (natural wetlands, wildfires, geologic emissions, wild animals (including termites) and marine $CH_4$ hydrates) contribute to the global $CH_4$ emissions (Kirschke et al., 2013). The main mechanism removing $CH_4$ from the atmosphere is the chemical reaction of $CH_4$ with OH radicals in the troposphere. Together with the other sink processes (such as stratospheric loss, reaction with Cl radicals in the marine boundary layer and soil uptake) the OH sink determines the recent atmospheric l̶i̶f̶e̶ ̶t̶i̶m̶e̶lifetime of $CH_4$, which is ̶b̶e̶t̶w̶e̶e̶n̶ ̶8̶.̶4̶ ̶a̶n̶d̶ ̶1̶0̶.̶6̶9.1 ± 0.9 years (N̶a̶i̶k̶Prather et al., 2̶0̶1̶3̶2012).

Despite the fact that the total $CH_4$ emissions today are well known, there is still considerable debate about the individual contributions of (in particular small) $CH_4$ ̶s̶o̶u̶r̶c̶e̶ ̶a̶n̶d̶ ̶t̶h̶e̶i̶r̶ ̶v̶a̶r̶i̶a̶b̶i̶l̶i̶t̶y̶.
[revised manuscript text omitted]